

# Low frequency magnetic variations at high-$\beta$ Earth bow shocks

Anatoli A. Petrukovich[1], Olga M. Chugunova[1], and Pavel I. Shustov[1]

[1]Space Research Institute of Russian Academy of Sciences, Moscow, Russia

**Correspondence:** A.A.Petrukovich (a.petrukovich@cosmos.ru)

**Abstract.** Earth's bow shock in high $\beta$ (ratio of thermal to magnetic pressure) solar wind environment is a rare phenomenon. However such an object is ubiquitous in astrophysical plasmas. Typical solar wind parameters related with high $\beta$ (here $\beta > 10$) are: low speed, high density and very low IMF 1–2 nT. These conditions are usually quite transient and need to be verified immediately upstream of the observed shock crossings. We survey statistics of high-$\beta$ shock observations by near-Earth spacecraft since 1995. About a hundred crossings were initially identified mostly with oblique or quasi-perpendicular geometry and high Mach number. In this report 22 crossings by Cluster project are studied with multipoint analysis, allowing to determine spatial scales. Observed shock structure is different from that for supercritical shocks with $\beta \sim 1$. The main magnetic field increase is smeared to couple tens of seconds and is dominated with magnetic variations $\sim$0.1–0.5 Hz (in some events — 1–2 Hz). Their polarization has no stable phase and is closer to linear, while spatial scales are of the order of hundred km at 0.1–0.5 Hz.

## 1 Introduction

Shocks are the primary dissipation mechanism in space plasmas with supersonic flows (Sagdeev, 1966; Kennel et al., 1985; Krasnoselskikh et al., 2013). A brand new branch of plasma science, theory of collisionless shocks, appeared in the sixties, in response to new space observations. Solar wind forms the bow shocks at planets and comets, as well as the termination shock at the heliospheric interface. Interplanetary shocks develop inside the heliosphere after solar eruptions, when large-scale transient structures propagate relative to the regular solar wind flow. In the distant space, shocks are associated with supernova explosions, stellar winds, collisions of galaxy clusters and are believed to have a leading role in the acceleration process of cosmic rays (Axford et al., 1977; Krymskii, 1977). Physics of space shocks was reviewed in AGU Geophysical Monographs, volumes 34 and 35 (1985). The Earth bow shock has been most thoroughly studied since the launch of the first spacecraft and is the main source of our in-situ knowledge of collisionless shock structure and dynamics.

Electromagnetic fields and waves in collisionless plasma shocks are of primary importance. Due to presence of magnetic field a wide variety of shock types exists with quite differing structure (Kennel et al., 1985). Magnetic field vector enters Rankine-Hugoniout equations, defining the relation between upstream and downstream conditions. In the absence of col-





lisions, kinetic mechanisms of field-particle interactions are responsible for dissipation and particle acceleration (Sagdeev, 1966; Krasnoselskikh et al., 2013). With quasi-perpendicular shock geometry (when the angle between the shock normal and the upstream magnetic field vector is closer to $90^o$ ) ions cannot escape upstream and relatively sharp shock transition forms with the ion-scale width (several thousand km). With quasi-parallel geometry (the angle is closer to $0^o$) ions easily escape

upstream along the magnetic field and the shock transition smears to the scales around several Earth radii (Scudder et al., 1986; Burgess et al., 2005). Oblique shocks (angles around $45^o$) are in a sense intermediate in properties, when ions partially are capable to escape upstream, but generally have rather spatially localized transition similar to quasi-perpendicular ones.

Besides these large-scale magnetic field structure also of interest at the Earth's bow shock are relatively low frequency magnetic variations (from one tenth to few Hz) with visually maximal amplitudes, which actually form the primary shock front

structure, dissipating ions. For example, in a supercritical quasi-perpendicular shock, the oblique whistler waves near the lower-hybrid frequency ($\sim$5 Hz) form the magnetic ramp via the non-linear steepening and decay cycle (Krasnoselskikh et al., 2002, and references therein). In the several studies the wavelength of these waves and the scale of shock ramp were determined to be around 10-s of km and oscillations were in fact identified as whistlers (Petrukovich et al., 1998; Walker et al., 2004; Hobara et al., 2010; Schwartz et al., 2011; Dimmock et al., 2013; Krasnoselskikh et al., 2013). Cyclic shock reformation is

typical also for quasi-parallel shocks with substructures known as SLAMS and oblique shocks (Lefebvre et al., 2009). The specifics of a plasma wave mode, driving the front reformation, depends of local plasma parameters, Mach number, etc. Immediately downstream of the shock front plasma waves at the frequencies below the ion cyclotron one were attributed to mirror, ion cyclotron, intermediate modes (e.g., Balikhin et al., 1997; Czaykowska et al., 2001). Yet one more issue of interest is electron heating. It requires sufficiently small scale variations for non-adiabatic acceleration and following isotropisation

(Balikhin et al., 1993; Vasko et al., 2018).

Of interest to several astrophysical applications are shocks in a weak magnetic field environment (high-$\beta$ shocks),common in interstellar and intergalaxy space (e.g., Markevitch and Vikhlinin, 2007; Donnert et al., 2018). $\beta$ is a dimensionless parameter, a ratio of plasma thermal to magnetic energy density. Unfortunately, observations of high $\beta$ shocks near Earth are quite rare, since the solar wind plasma usually has $\beta \sim$1. Very few such investigations were published, merely checking

validity of Rankine-Hugoniot conditions and marking very high amplitude of magnetic variations (Formisano et al., 1975; Winterhalter and Kivelson, 1988; Farris et al., 1992). Note, that is some investigations moderate $\beta \geq 1$ was termed as "high-$\beta$" regime (e.g., $\beta = 2.4$ in Scudder et al., 1986).

We perform an extended experimental study of high-$\beta$ bow shock with a goal to better understand essential plasma processes and instabilities driving shock front formation and plasma dissipation. We scanned 1995–2017 observations by Interball (1995–

2000), Geotail (since 1995), Cluster (since 2000) and THEMIS (since 2007). We use $\beta >$10 criterion, which is justified in the course of this presentation. In this initial investigation we present a first, to the best of our knowledge, multi-point analysis of dominating low-frequency magnetic varations at high-$\beta$ shock transition using observations of Cluster project. We also present the occurrence of high-$\beta$ solar wind. Though such solar wind statistics is generally known (review in Wilson et al., 2018), some issues relevant to shock identification and analysis are still worth addressing.




For initial selection we use orbital and spin-averaged magnetic field data from CDAWeb archive. For the detailed analysis we used full-resolution Cluster FGM magnetic field (here with the sampling ∼20 Hz) (Balogh et al., 2001) and HIA/CODIF ion data (sampling once in 8 seconds) (Rème et al., 2001) from Cluster Science Archive. Solar wind and IMF data were taken from OMNI-2 data set, the 1-hour variant was used for the initial survey, and the 1-min variant — for the final categorization of crossings. $\beta$ values are precalculated in OMNI-2, assuming constant electron temperature, He++ fraction and temperature. To access possible solar wind variability we use also ACE and Wind final Earth-shifted data from OMNI archive. All vectors are in GSE frame of reference.

## 2   Solar wind statistics and details of search procedure

We use 1-hour OMNI data for the period 1995–2017 to determine the occurrence of high $\beta$ solar wind for the subsequent shock analysis. The average solar wind $\beta$ is somewhat large than unity. High $\beta$ conditions are unevenly distributed across solar cycles (Fig. 1), being more frequent at the solar minima 1996–1997 and 2007–2009. For the threshold $\beta > 10$ there are 50–500 hours per year, while for $\beta > 20$, the number is about 3–5 times smaller.

Figure 2 shows distributions of magnetic field magnitude, solar wind speed, density and total static pressure for the full dataset of one-hour values during 1995–2017 and for the subset $\beta > 10$. The high $\beta$ corresponds to slow, cold and dense solar wind with low magnetic field (ion temperature not shown here). However the total static (magnetic plus thermal) pressure distribution is similar (Fig. 2b). Thus the high-$\beta$ events are mostly depressions of magnetic field, compensated (at least on average) by increase of plasma density. The only notable difference of distributions for $\beta > 20$ (Fig. 2a, red line) is more frequent presence of magnetic field ∼1 nT, with the average 1.6 nT, while for $\beta > 10$ the average is ∼2.2 nT.

More than 50% of events with $\beta > 10$ have one-hour duration (one point in the analyzed OMNI variant, not shown here). A sample event is in Fig. 3. There is about one-hour long decrease of magnetic field and density increase, corresponding to $\beta$ ∼20. At an occasional depletion of magnetic field below 2 nT $\beta$ jumps to about 40–80 for few minutes. Since formation of high $\beta$ conditions mostly depends on subtle variations of magnetic field magnitude around 1–2 nT (note, that $\beta$ has square dependence on magnetic field), it should be quite sensitive to spatial inhomogeneity of solar wind and IMF, and, in particular, to differences between those detected at L1 (in OMNI dataset) and actually hitting Earth. Fig. 5 shows comparison of $\beta$ calculation for Wind and ACE 1-hour data (only for times, when Wind data were used in OMNI). The scatter is indeed large.

We formulate several conclusions important for our specific shock analysis. (1). Solar wind intervals with high $\beta = 10$–20 are rare, but not extremely rare, and occur mostly during solar minimum. Thus some spacecraft (or the project phases with the specific orbit or spacecraft separation) may almost completely miss such events. (2). Duration of intervals of interest is relatively short, thus selection of shocks with stable upstream conditions may be not always possible. (3). Very low interplanetary magnetic field, necessary for high-$\beta$ events, is subjected to strong (in relative terms) intrinsic spatial and temporal variability, thus actual $\beta$ conditions and IMF vector need to be always rechecked with local measurements. This issue is further illustrated with the event selection results below and is elaborated more in Discussion.





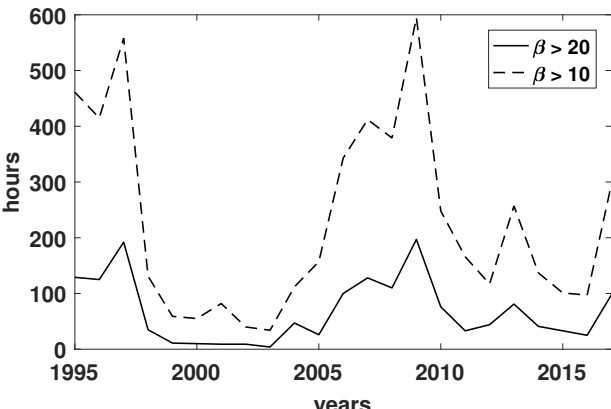

**Figure 1.** Number of hours with high $\beta$ with respect to calendar year.

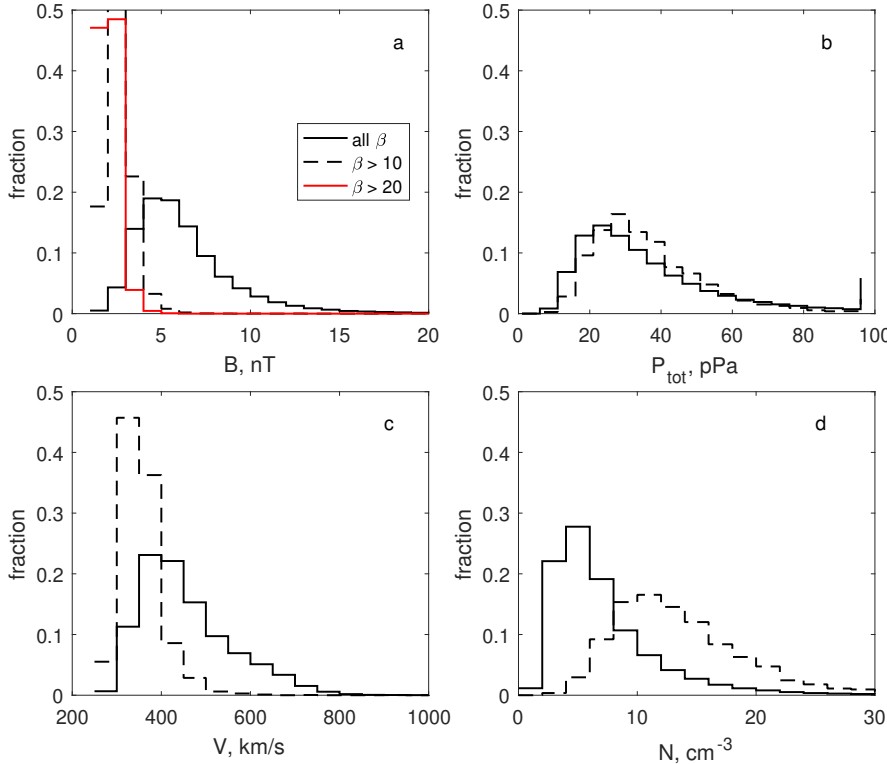

**Figure 2.** Histograms of solar wind and IMF occurrence for 1995–2017 (solid lines) and for $\beta > 10$ (dashed lines) subset. (a) Total magnetic field (red line corresponds to $\beta > 20$) , (b) total static pressure, (c) solar wind speed, (d) ion density.

Since the high-$\beta$ shocks are rare, it is unreasonable to search for them, rechecking every registered event. It is more practical first to identify the intervals with the suitable conditions of solar wind. The semi-automated algorithm is used to assemble





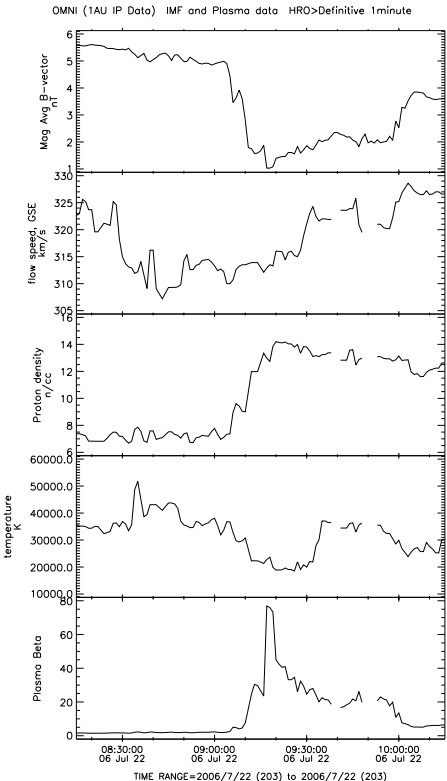

**Figure 3.** Example of high-$\beta$ interval. From top to down: magnetic field magnitude, solar wind speed, proton density, proton temperature, plasma $\beta$. 1-min OMNI data set used.

initial statistics of the shock candidates. For each 1-hour point in OMNI with $\beta > 10$, we check for possible spacecraft location within 5 $R_E$ from the model bow shock (Farris et al., 1991). In a case any spacecraft is in the right place, the plots of solar wind, IMF, local magnetic field and plasma parameters are analyzed visually in the 5-hour window around the selected hour.

5   These broad temporal and spatial spans are used to ensure that all possible crossings of a moving bow shock are captured for future analysis. Only events with the clear shock traversals (jumps in magnetic field and ion density) are accepted. Such a manual selection has definite bias to quasi-perpendicular and oblique shocks (which usually have a step-like appearance), but it is considered acceptable for this particular study. The most of these initially selected intervals actually contain no shock crossings.

10  Discovered particular shock crossings are checked with 1-min OMNI data. Plasma $\beta$ is often below 10, either because registered shocks are just outside of initially selected hours, or because $\beta$ varied on a time scale, smaller than an hour. Since a change of $\beta$ is usually related with the solar wind density change, it is associated also with the dynamic pressure change.



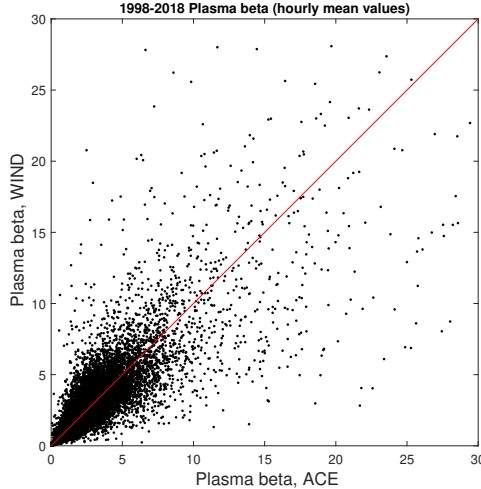

**Figure 4.** Comparison of Wind and ACE $\beta$ using 1-hour data. See text for details. Red line is bisector

The latter drives a large-scale shock motion and probability of shock registration by a spacecraft increases. In fact, many shock crossings are registered at a boundary of $\beta$ change and such events are also discarded, since it isimpossible to attribute them to stable upstream plasma conditions.

Finally the list contained about a hundred individual crossings with average $\beta$ about 20 (taken as 1-min OMNI value at the moment of shock front crossing). About ten events occurred with very high $\beta > 40$. The choice of initial threshold $\beta > 10$ (for 1-hour points) was finally justified at this stage, since a variant with initial $\beta > 20$ resulted with the almost empty final list. However, all these events still need a more detailed confirmation, in particular, of local high $\beta$, stable enough crossing velocity, plasma data availability etc.

For the specific multipoint analysis in this investigation we selected 22 verified Cluster project crossings with relatively small spacecraft separation. One event is from 2003, with the Cluster tetrahedron size of about 300 km, while the other are for the late years 2008–2016, when separation only between a pair of Cluster spacecraft C3 and C4 was controlled (30–150 km for our events). The full list is in Table S1 in Supplement 1. This uneven annual distribution is a consequence of the solar cycle dependence (Fig. 1). Events are grouped in 7 days. Specifically, 5 crossings are registered within one hour at December 18, 2011, 4 crossings — within two hours at January 3, 2008, 8 crossings — within two hours at January 4, 2008, 2 crossings — within one hour at February 16, 2012. However not all these adjacent crossings are similar. Some of these examples are presented below.



## 3 Shock examples

### 3.1 Event 1

The first example is registered by Cluster C3 and C4 spacecraft on 18 December 2011 (14:36–14:40 UT) with the separation
36 km. The spacecraft orbit is almost parallel to the model shock front (Fig. 5), but shock velocity is definitely much higher
than the spacecraft velocity. Solar wind speed is small $\sim$260 km/s, IMF magnitude — 2.5 nT (all characteristics are in Table
S1). Alfven Mach number is $\approx$18, magnetosonic Mach number is $\approx$5, $\beta$ (according to 1-min OMNI) is 10.8. Solar wind
magnetic field measured locally by Cluster is the same as OMNI data (compare two lines in Fig. 6d), therefore OMNI $\beta$
value is confirmed. OMNI IMF vector direction is only $\sim$10$^o$ different with the local upstream field taken at 14:40–14:41
UT (not shown here). The model shock normal angle with respect to OMNI (local) IMF $\theta_{Bn}$ is 46$^o$ (54$^o$) (using Farris et al.
(1991) model). The coplanarity calculation for the shock normal results in $\theta_{Bn}$ equal to 42$^o$. Thus this is quasi-perpendicular
or oblique supercritical bow shock with the reliably determined geometry. It's structure for more standard $\beta$ is well studied
(Scudder et al., 1986; Krasnoselskikh et al., 2013; Lefebvre et al., 2009).

Fig. 6 contains overview of magnetic field and plasma parameters. The transition lasts about 200 seconds 14:37:00–14:40:30
from the first signs of gyrating ions upstream (Fig. 6f) up to the stable downstream conditions. The increase in magnetic field
magnitude (aka shock ramp in a quasi-perpendicular case) is smeared within half a minute 14:37:45-14:38:20 UT  and is
accompanied with the similar smeared increase of ion density. The nominal shock front transition is somewhat arbitrarily
placed at 14:37:45 UT (marked by vertical line) at a first extended peak of magnetic field. The magnetic field increase is wavy,
rather than regular or step-like, magnetic magnitude immediately downstream is often down to 5 nT. Thus it proved to be
impossible to determine with multipoint analysis the shock speed. The final value of downstream magnetic field is around 10
nT, and compression ratio is thus close to maximally possible value of 4, in accordance with the high Mach number.

Despite the described smeared magnetic field increase, the full shock transition is rather compact and coherent and thus it
is distinctly different from what expected for quasi-parallel shock with multiple shocklets (Burgess et al., 2005). The smeared
increase of magnetic field magnitude may be attributed to relatively large ion gyroradius in low IMF, or interpreted as a cyclic
reformation, similar to that of oblique shock (Lefebvre et al., 2009). A more detailed phenomenological description of this
shock transition requires analysis of ion kinetics, which will be performed elsewhere.

We highlight in Figure 7 the interval with the strongest low-frequency magnetic variations. Frequency spectra are in Figure 8.
The magnetic profile is dominated by a variation with frequency around 0.3 Hz and amplitude up to 20 nT, more pronounced
in $B_y$. An interval 14:37:27–14:37:47 is taken to estimate the wavelength. The main oscillation (0.3 Hz) is very similar at
two spacecraft and visually the time shift between C3 and C4 is about a fraction of a second. Since the variation has a clear
dominating frequency it is more convenient to perform the time-domain multi-point analysis.

The parameters of magnetic variations, filtered in frequency range 0.1–0.77 Hz, are in Table 1. Vector of maximum variance
is almost along local magnetic field ($B_y$ component dominates), of minimum variance — along $Z$. Ratios of eigenvalues are
$\lambda_{min}/\lambda_{int} = 0.34$, $\lambda_{int}/\lambda_{max} = 0.58$, and one may assume elliptic polarisation. The time shift between magnetic measure-
ments along the maximum variance component, determined with the correlation analysis, is 0.13 s. This value is rather reliably




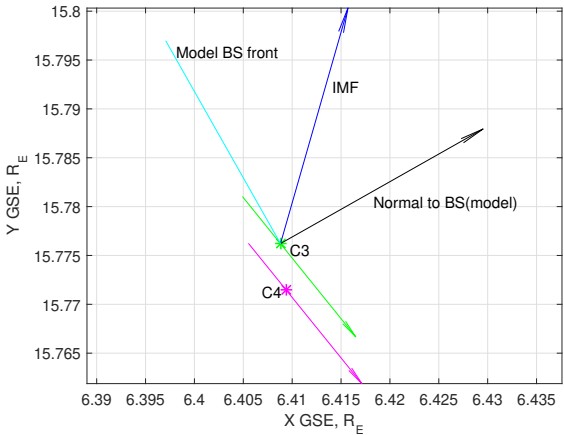

**Figure 5.** Spacecraft orbit and model shock position for shock 12 December 2011.

calculated, since it is 2–3 times larger than the sampling interval. The spacecraft separation along the minimum variance direction is 10 km and the resulting wavelength estimate is ∼250 km. However the hodograph of magnetic field rotation (Fig. 9) shows that the polarization actually might be linear with the variable eigenvector (but mostly along the determined maximum variance). In such a case propagation direction cannot be defined with the variance analysis. Alternatively, for the compressive low frequency MHD waves the propagation direction can be determined with the coplanarity approach (Hubert et al., 1998) (the maximum variance direction, the magnetic field direction and the wavevector should be in the same plane). However, in our case, the angle between the maximum variance direction and the local magnetic field is rather small (only $12^o$) and coplanarity calculation result would be unrealiable.

We also estimate the span of principally possible wavelengths. The maximal one is ∼900 km, obtained taking full spacecraft separation 36 km. The Doppler shift is 0.04–0.58 Hz, depending on a wavelength and taken local proton velocity value (full 146 km/s or its projection to minimal eigenvector 41 km/s).

Finally we note the oscillations with higher frequency about 1 Hz and smaller amplitude of couple nT, which are best observable in $B_z$ component (Fig. 7c and Fig. 8). The eigenvalue ratios (after filtering the frequency range 0.7–10 Hz) are $\lambda_{min}/\lambda_{int} = 0.68$, $\lambda_{int}/\lambda_{max} = 0.49$, thus reliable determination of any wave proper direction is definitely not possible. Oscillations are quite different at two spacecraft and the multipoint analysis also proved to be not possible.



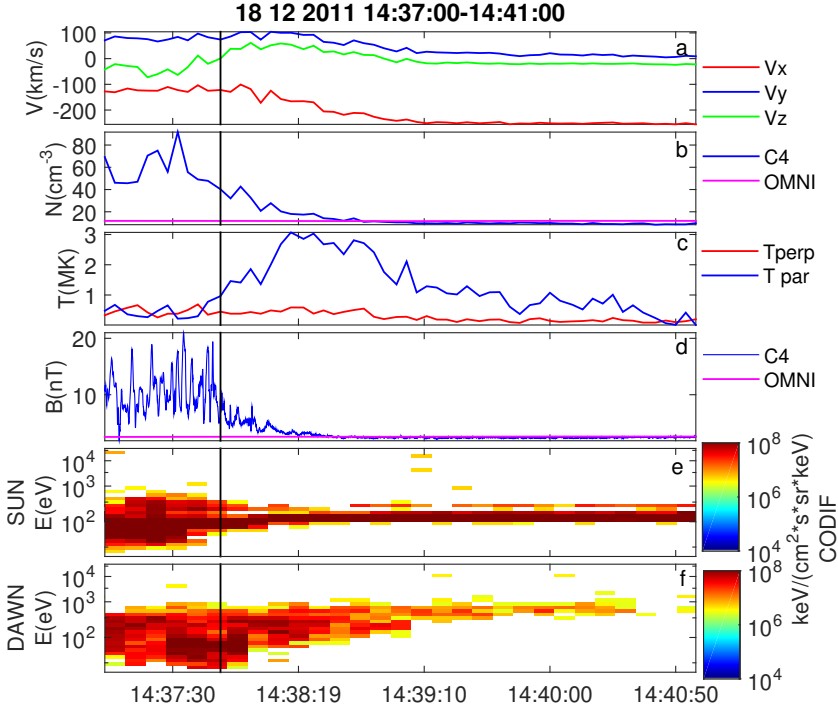

**Figure 6.** Overview of C4 magnetic and plasma measurements for event 18 December 2011. (a) proton velocity, (b) proton density and OMNI solar wind density, (c) proton parallel and perpendicular temperature, (d) magnetic field magnitude and OMNI IMF magnitude, (e,f) proton spectrograms for the sunward and dawnward looking sectors.

## 3.2 Event 2

A shock from January 4th, 2008 (16:00–16:04 UT) was registered with Cluster C3 and C4 separation about 40 km. General
20  event parameters are in Table S1, overview of plasma and magnetic field parameters is Fig.S1 in Supplement. The detailed
wave activity at the front is presented in Fig. 10. Solar wind parameters and general crossing structure are very similar to that
for the Event 1. Solar wind speed is small ∼315 km/s, IMF magnitude — 2.4 nT. Alfven Mach number is ≈23, magnetosonic
Mach number is ≈7, current $\beta$ (according to 1-min OMNI) is 12.2. Solar wind magnetic field measured locally by Cluster
is the same as OMNI data (compare two lines in Fig. S1d), therefore OMNI $\beta$ value is confirmed. All variants for $\theta_{Bn}$ give
25  around $40^o$.

The transition lasts about 2 minutes 16:00:50-16:02:50 from the first signs of gyrating ions upstream to stable downstream
conditions (Fig. S1f). The jump in magnetic field magnitude and ion density is smeared within half a minute 16:01:30-16:02:00
UT, and is wavy rather than step-like, downstream magnetic magnitude is often as small as 2–5 nT. The nominal shock front
transition is somewhat arbitrarily placed at 16:01:35 UT at a first extended peak of magnetic field.





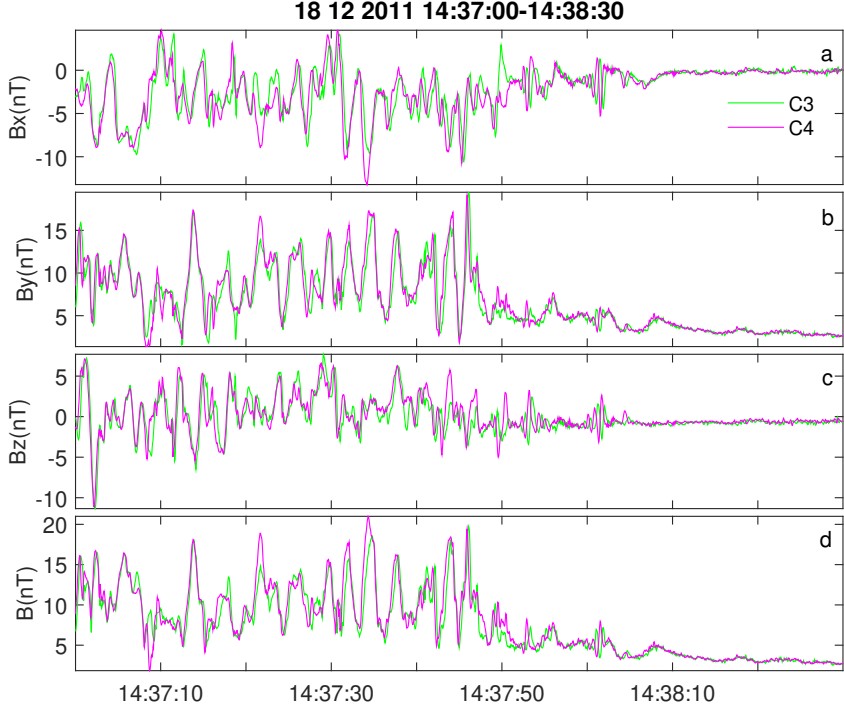

**Figure 7.** Full resolution magnetic waveform for shock 18 December 2011. In panels (a-d) are components and total value of magnetic field.

The full resolution waveform is in Figure 10. Similar to Event 1, there is a dominating oscillation with frequency about 0.4–0.5 Hz, as well as the lower amplitude waves with frequency above 1 Hz (Fig. S2). The specific feature is strong difference of C3 and C4 oscillation amplitudes during the first 20 s downstream the front (16:01:25–16:01:35 UT) despite relatively small separation. The presence of such difference in amplitudes is typical for all shocks registered during this day (8 crossings within 2 hours in Table S1).

It is possible to perform multipoint separation analysis for the interval 16:01:15-16:01:25, where two waveforms in $B_y$ component (Fig. 10b) are very similar and shifted by a fraction of period. All wave parameters (filtered in the range 0.1–2 Hz) are in Table 2. As in Event 1, maximum eigenvector is almost along $Y$, medium eigenvector — along $X$. Ratios of eigenvalues are $\lambda_{min}/\lambda_{int} = 0.15$, $\lambda_{int}/\lambda_{max} \approx 0.5$, thus minimum variance (nominal propagation) direction is well defined. The time shift between the magnetic measurements along the maximum variance component is 0.22 s (determined with correlation analysis), while the spacecraft separation along the minimum variance direction is 6.8 km. The resulting wavelength estimate is 61 km for the peak frequency 0.5 Hz. This value is close to spacecraft separation distance (about 40 km) and thus is generally consistent with the nearby observation of substantial difference between magnetic fields at C3 and C4 (at 16:01:25–16:01:35).



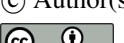

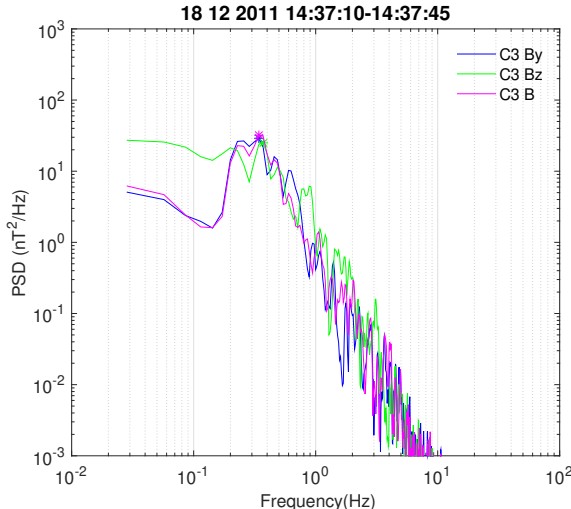

**Figure 8.** C3 frequency spectra for $B_y$, $B_z$ components and magnetic field magnitude for shock 18 December 2011.

**Table 1.** Wave analysis data for shock 18 December 2011, 14:37:27–14:37:47.

| | |
|---|---|
| max eigenvector, $V_{max}$ | -0.23, 0.94, 0.27 |
| med eigenvector, $V_{med}$ | 0.97, 0.20, 0.15 |
| min eigenvector, $V_{min}$ | -0.08, -0.29, 0.95 |
| eigenvalues | 2.23, 6.64, 11.50 |
| magnetic field C3, $B_3$ (nT) | -3.58, 9.53, 0.96 |
| local proton velocity C4 (km/s) | -118.1, 82.1, -29.29 |
| angle, $V_{max}$ and IMF | $34^o$ |
| angle, $V_{min}$ and IMF | $110^o$ |
| angle, $V_{max}$ and $B_3$ | $12^o$ |
| angle, $V_{min}$ and $B_3$ | $99^o$ |
| peak frequency in max component | 0.3 Hz |
| time shift in magnetic field along $V_{max}$ | 0.13 s |
| separation along $V_{min}$ | 10 km |
| wavelength | 252 km |

The hodograph of magnetic field rotation (Fig. 11), however, shows absence of any stable polarization, which can be interpreted as sometimes linear, sometimes circular. The coplanarity approach again can not be used here to confirm the wave



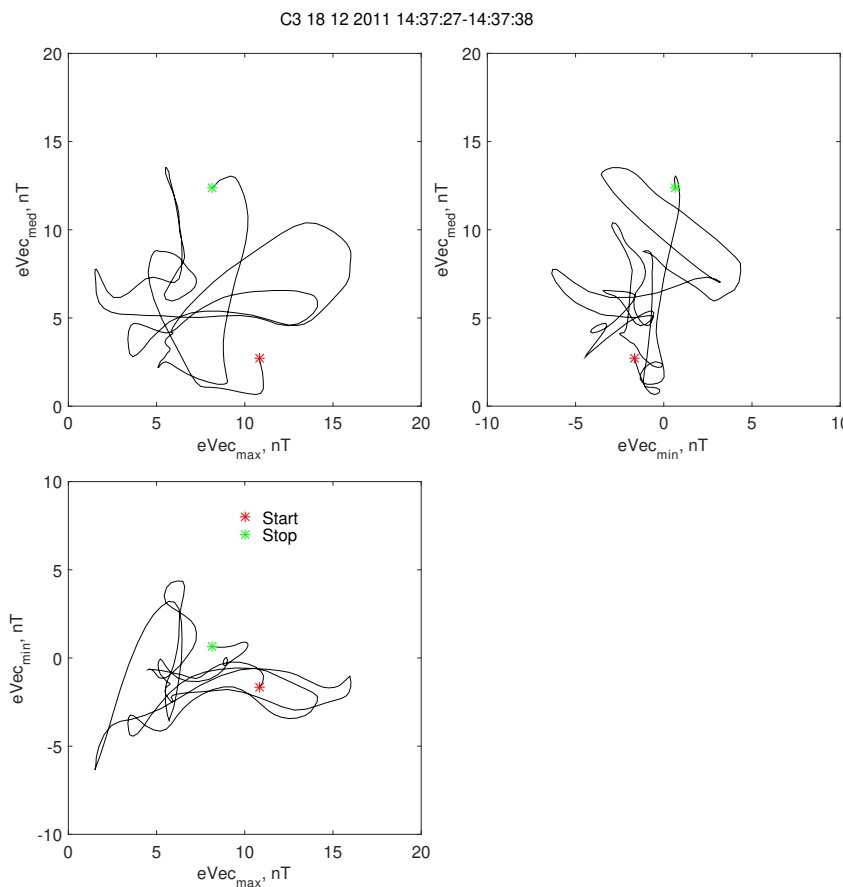

**Figure 9.** Hodographs of C3 magnetic field in eigenvector coordinates for shock 18 December 2011.

vector direction since the angle between the maximum variance direction and the local magnetic field is rather small ($20^o$). The maximum possible wavelength (if spacecraft separation along wave vector is maximal 40 km) is ∼400 km.

### 3.3 Event 3

One more crossing is from January 3rd, 2008 (14:30–1435 UT) with Cluster separation ∼100 km (Table S1, Fig. S3 in Supplement). OMNI data showed very low IMF (1.1 nT) and $\beta = 39$. Solar wind speed is small ∼321 km/s, Alfven Mach number

5  is ≈42, magnetosonic Mach number is ≈7. The model $\theta_{Bn}$ is $47^o$. In Fig. 12 we present a view of local magnetic field along with OMNI data. Though local upstream magnitude is approximately equal to that in OMNI (except starting from 14:30 UT closer to the shock), the upstream field direction is changing by more than $90^o$ and the model $\theta_{Bn}$ is also changing to more





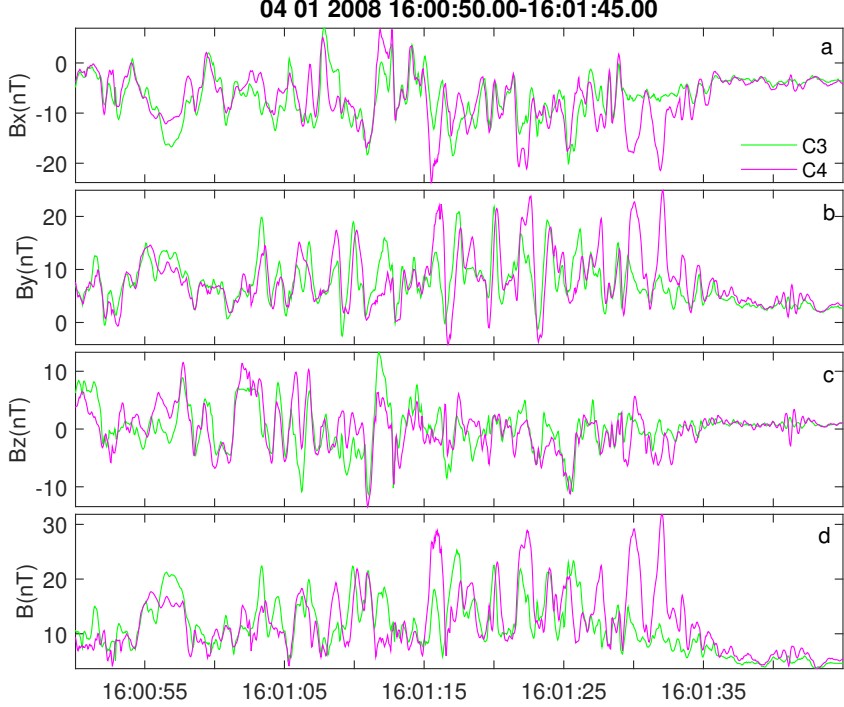

**Figure 10.** Full resolution magnetic waveform for shock 04 January 2008. In panels (a-d) are components and total value of magnetic field

perpendicular geometry. The presence of an earlier shock crossing at 14:20 UT may also affect observed upstream conditions. Downstream magnetic field is also strongly changing direction with a temporal scale of about a minute (Fig.12, right side). Therefore for this shock reliable determination of magnetic geometry is impossible. This problem may be inherently related with very small value of upstream magnetic field.

Fig. S3 contains overview of magnetic field and plasma parameters. The transition lasts about 2.5 minutes 14:32:00–14:34:30 from the first signs of gyrating ions upstream and growth of parallel ion temperature (Fig. S3e,f) to stable downstream conditions. The jump in magnetic field magnitude is smeared within half a minute 14:34:00–14:34:30 UT, It is wavy rather than step-like and magnetic magnitude downstream is often as small as 1–2 nT. The nominal shock front transition is somewhat arbitrarily placed at 14:34:10 UT (marked by vertical line in Fig.S3). Some increase of variation amplitudes around 14:34:10 can be interpreted as a localized front intensification or as a result of shock bounce motion.

The detailed view of magnetic variations is in Fig. 13. Only relatively high frequency oscillations about 2 Hz are present (frequency spectra are in Fig. S4). There are no wave packets with the stable phase. For example, at 14:34:10–14:34:14 UT $X$ and $Z$ components are in anticorrelation for C3 and C4, while immediately near, at 14:34:08–14:34:10 UT these components are in phase. Therefore, the reliable multipoint analysis for this event is impossible. Magnetic field hodograph plot for 14:34:10–



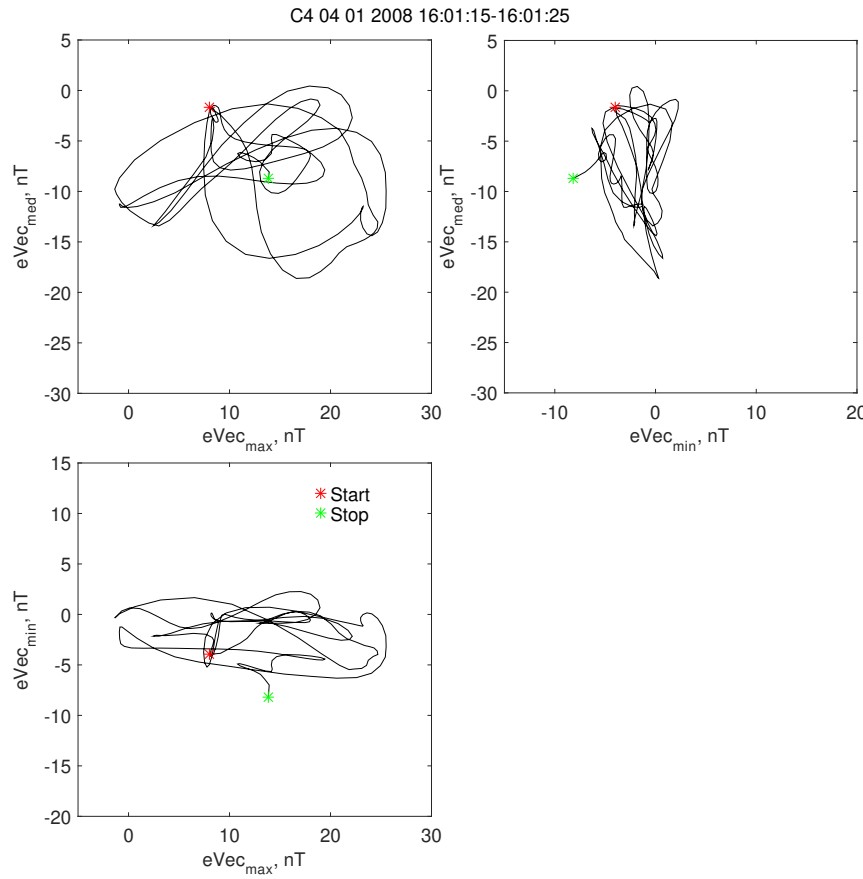

**Figure 11.** Hodographs of C4 magnetic field in eigenvector coordinates for shock 04 January 2008.

14:34:14 is in Fig.14. It confirms unstable (but consistent with the changing linear) polarization. However, assuming that C3 and C4 variations are mostly in antiphase (half a period between spacecraft), one gets the maximal wavelength estimate ~200 km.

### 3.4 Observation summary and statistics

Our statistics includes 22 oblique and quasi-perpendicular shocks. Three examples well illustrate typical shock properties. The minimum $\theta_{Bn}$ $37^o$, two largest ones are $62^o$ and $83^o$. Values of $\beta$ range from 39 to 7.5. All cases are supercritical shocks with magnetosonic Mach number more than 5.5. Alfvenic Mach numbers are large because of large $\beta$. All shocks exhibit a clear several-minute-long transition zone between solar wind ion flow and magnetosheath. The main increase of magnetic field and ion density has duration about several tens of seconds. The observed shocks, as concerns their general structure, are typical




**Table 2.** Wave analysis data for shock 04 January 2008, 16:01:15–16:01:25.

| | |
|---|---|
| max eigenvector, $V_{max}$ | -0.46 0.87 0.17 |
| med eigenvector, $V_{med}$ | 0.88 0.42 0.22 |
| min eigenvector, $V_{min}$ | -0.12 -0.25 0.96 |
| eigenvalues | 3.4, 22.9, 45.3 |
| magnetic field C3, $B_3$ (nT) | -9.05, 9.85, -0.75 |
| local proton velocity C4 (km/s) | -178.3, 125.7, -67.4 |
| angle, $V_{max}$ and IMF | $46^o$ |
| angle, $V_{min}$ and IMF | $79^o$ |
| angle, $V_{max}$ and $B_3$ | $20^o$ |
| angle, $V_{min}$ and $B_3$ | $99^o$ |
| peak frequency in max component | 0.5 Hz |
| time shift in magnetic field along $V_{max}$ | 0.22 s |
| separation along $V_{min}$ | 6.8 km |
| wavelength | 61 km |

oblique/quasi-perpendicular shocks, with the somewhat smeared main magnetic field increase. This magnetic profile is typical for all our shocks irrespective of $\theta_{Bn}$ angle.

10   On a smaller time scale of seconds, the magnetic profile is dominated by very large amplitude magnetic variations, gradually growing in the course of magnetic field increase towards downstream. As a result, the exact location of the 'main' magnetic jump (aka ramp for supercritical quasi-perpendicular shocks) can not be defined. This presence of high-amplitude variations is in agreement with the previous publications (Winterhalter and Kivelson, 1988).

The three examples show characteristics of the dominating magnetic variations, typical for all considered events. The detailed multipoint variation analysis allowed to obtain following new information. In the most of shocks (and in Examples 1 and 2) the variations exhibit the well defined frequency peak ∼0.2–0.5 Hz. The magnetic phase portrait of these variations is irregular, with no clear persistent polarization. It can be also interpreted as a linear polarization with the frequently changing main

5   direction. Such polarization does not allow to determine reliably the wave propagation direction and the wavelength. We get the estimates only in the range several-tens-hundreds km.

Two shock events (Dec. 31, 2003 and our Example 3, Jan. 3, 2008 14:32 UT)) have dominating ∼2 Hz variations, visually with more harmonic waveform, but also with the unstable phase. These two shocks are not different from the other events in terms of their general parameters. Moreover one of them (Event 3 above) is registered just 10 min after another crossing,

10   which exhibited the first type of variations. Therefore the presence of '2-Hz' waves might be due to some temporal shock front evolution.





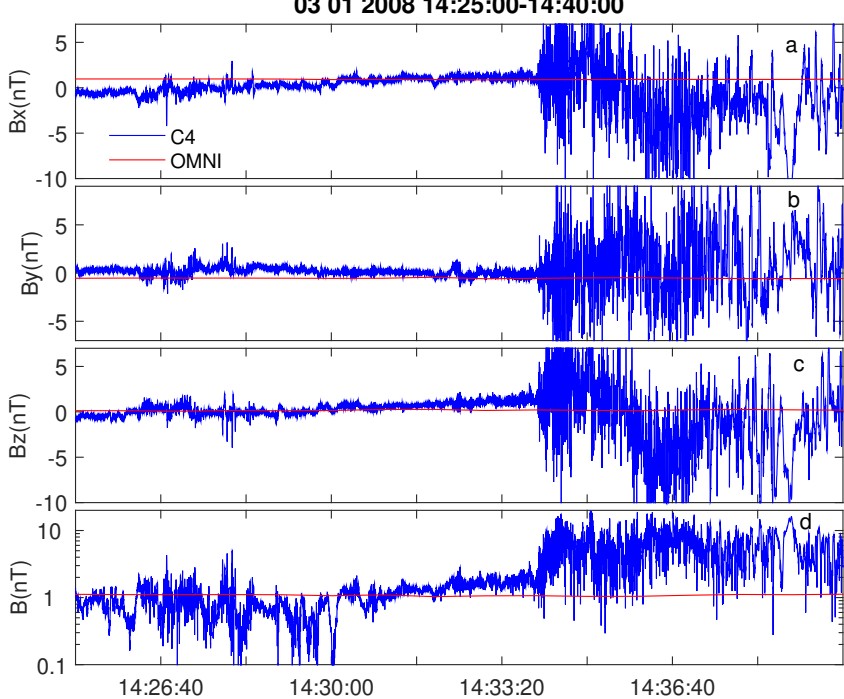

**Figure 12.** Local upstream and OMNI magnetic field for shock 03 January 2008. In panels (a-d) are components and total value of magnetic field

## 4 Discussion

### 4.1 Reliability of solar wind input

High-$\beta$ solar wind is relatively rare at the Earth orbit. In our study we accepted somewhat ad-hoc threshold of high $\beta$ equal to
10. Such interplanetary conditions tend to occur during solar minima, being created by slow cold dense solar wind with low
IMF (1–2 nT). However is not always easy to confirm that the observed shock crossing actually occurred in high-$\beta$ solar wind
interval, identified in OMNI. The first set of problems is related with association of particular crossings with the stable high $\beta$.
These problems are relatively straightforward to identify. A more substantial problem is related with the inherent solar wind
and IMF variability. We measure solar wind in L1 halo orbit, 1.5 million kilometers away from Earth and with halo radius not
less than 200 000 km (for ACE spacecraft). A substantial part of modern OMNI data are taken from Wind spacecraft, which is
currently on a much wider halo orbit (300–400 thousand km) (Podladchikova et al., 2018). Solar wind and IMF structures at L1
are not necessarily the same, that actually affect the magnetosphere. The most questionable is spatial persistence of relatively
small changes of IMF from 2 to 1 nT, required for creation of very high-$\beta$ intervals.





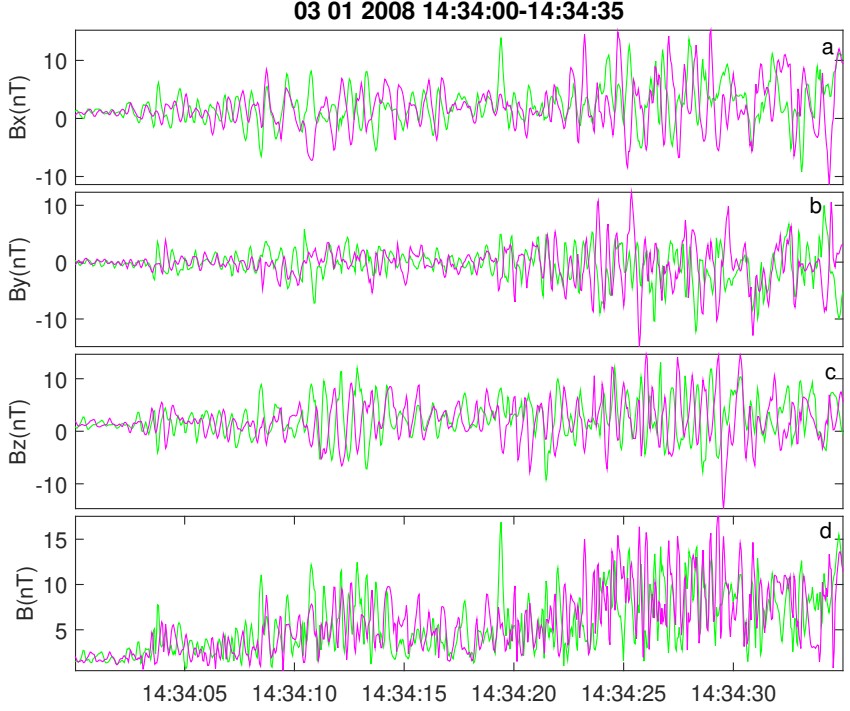

**Figure 13.** Full resolution magnetic waveform for shock 03 January 2008. In panels (a-d) are components and total value of magnetic field

Though the specific analysis of the spatial scales of high-$\beta$ areas in the solar wind was not performed, available reports indicate significant potential problems. The ISEE data study suggested that during periods of medium to low variance of magnetic field, magnetic features with scale widths of 20 $R_E$ perpendicular to the IMF may occur (Crooker et al., 1982). Comparison of L1 Wind and near-Earth Interball data for 1996–1999 have shown (Petrukovich et al., 2001), that the IMF structures, associated with geomagnetic storms (with the threshold of IMF $B_z$ GSM below –10 nT during 3 hours) are practically the same in L1 and the near-Earth orbits. However, about 20–80% of the smaller everyday IMF variations, causing substorms (several nT in magnitude on one-hour scale) are different by more than 25% .

Thus the applicability of very high $\beta$ values in OMNI to a shock study is not automatic. It is not always possible to check solar wind $\beta$ immediately before shock crossing. A spacecraft needs to probe pristine solar wind and then rapidly cross the shock, or there should be an additional near-Earth solar wind monitor. Magnetic field can be reliably measured with magnetometer (still assuming offset uncertainty of about 0.1 nT). Accuracy of ion density and temperature measurements is more problematic, since at L1 the specialized thoroughly calibrated instruments are used, while with a magnetospheric spacecraft, calibration could be rougher for the specific case of solar wind flow. Assumptions on the helium content and electron temperature, used





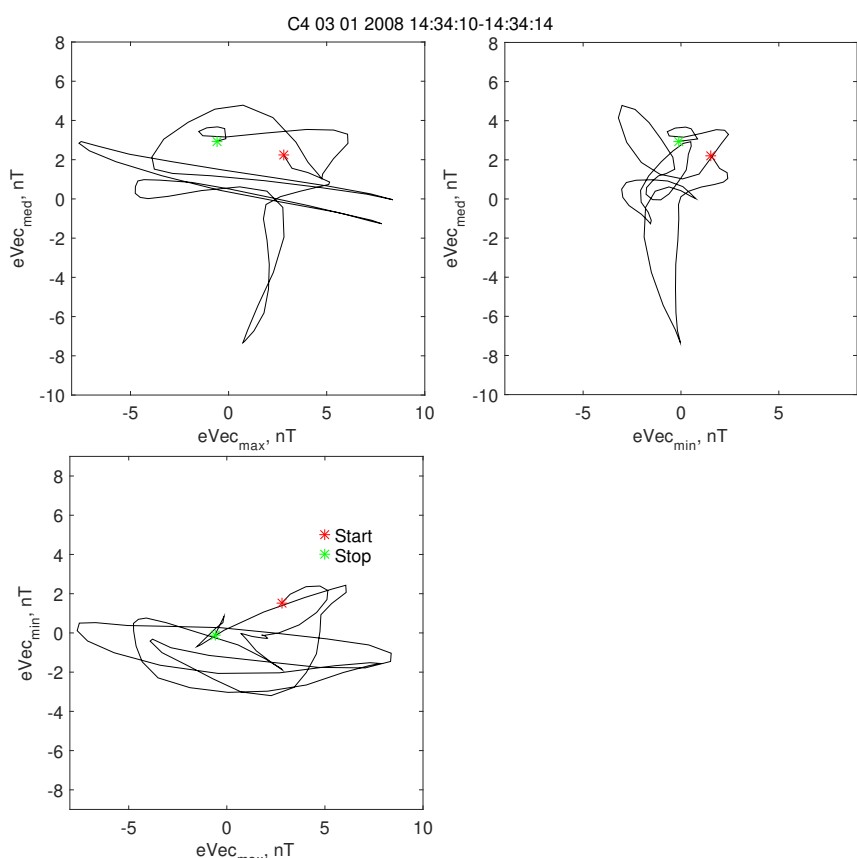

**Figure 14.** Hodographs of C4 magnetic field in eigenvector coordinates for shock 03 January 2008 for 14:34:10-14:34:14.



while OMNI $\beta$ calculations, may also result in some errors. Of course as a side product of such variability, an additional (relative to those found in the OMNI set) high $\beta$ intervals may actually form near the bow shock.

## 4.2 Shock properties

The rather compact large scale structure of the observed shock transitions is similar to that reported for oblique and quasi perpendicular shocks. It is distinctly different from the structure of quasi-parallel shocks, which are extended up to several Earth radii. However, there are some differences with low-$\beta$ shocks on a smaller scale. Duration of the main magnetic and ion density increase is several tens of seconds. Magnetic variations appear during this increase and have the amplitude comparable or larger than the background magnetic field, so that there is no 'stable' magnetic structure on the time scale of seconds. In comparison, for the supercritical quasi-perpendicular low-$\beta$ shocks, one usually defines, starting from the upstream, the prolonged interval of somewhat enhanced density and magnetic field (shock foot, lasting tens of seconds) and the sharp main increase (ramp, lasting seconds). The ramp is often used to determine the shock motion with multipoint measurements, but in our case it is impossible. The increased width of main magnetic jump and its wavy nature might be related with some essential scales in the high-$\beta$ plasma or be a sign of the unstable reforming shock front (e.g., Lefebvre et al., 2009).

A shock example with very low upstream magnetic field about $1\,\mathrm{nT}$ exhibits very variable direction of magnetic field both upstream and downstream, complicating definition of shock magnetic geometry. This issue might be of a special interest, since at some (very small) value of magnetic field, it's direction should become unimportant for the shock structure. In particular, there will be no difference between perpendicular and parallel shocks and the shock spatial scale should be defined with some nonmagnetic parameters. Ion escape upstream would be controlled then by some diffusive processes. Thus, one of the main topics for the future studies is to observe in greater detail the dependence of the shock scales on the value of $\beta$, especially for the events with the extreme $\beta$. Our statistics has about 10 events with $\beta$ in the range 40–100, comparable with that expected to the galaxy clusters plasma (Donnert et al., 2018).

## 4.3 Variation properties

Observed properties of low-frequency magnetic variations (linear polarization with very high amplitude, substantially changing the total magnetic field) immediately suggest their compressive nature and a strong spatial localization due to absence of any stable several-periods-long wave packets. Thus observed variations are strongly different from that in low-$\beta$ supercritical events (e.g. Krasnoselskikh et al., 2013), where clear whistler wave packets with elliptic polarization dominate. Observed polarization is also not consistent with the earlier suggested alfven mode (Kennel and Sagdeev, 1967).

Dominating wave mode downstream of the shock front was also addressed in a number of other investigations. Hubert et al. (1989) identified mirror waves comparing magnetic field with fast electron measurements of ISEE project. Balikhin et al. (1997) identified intermediate mode with two-point AMPTE data analysis. Lacombe et al. (1992) suggested for higher-$\beta$ shocks the mirror mode with linear polarization, and successfully used coplanarity assumption to define the wavevector direction. Czaykowska et al. (2001) have shown mostly compressive mirror mode in shocks with $\beta > 1$. Therefore, almost full variety of possible wave mode variants was identified.

However all these studies used several-minute data intervals, often several minutes away from the shock transition (and the highest amplitude waves at it), with the natural motivation to access the long sets of uniform variations. In the most of cases, the

analyzed frequencies were below 0.1 Hz. This approach is different from ours, in which we addressed relatively short intervals of most powerful oscillations. Also all variants of the plasma mode analysis critically depend on reliable determination of the wave propagation (wavevector) direction. It can be done with minimum variance analysis in the case of elliptic polarization or with the coplanarity supposition. Unfortunately in our case it proved to be impossible to determine the wavevector direction reliably by both methods. It should be noted that a linearly polarized variation with the high amplitude comparable or larger than

the background field inevitably has the maximum amplitude direction almost parallel to the main magnetic field, precluding the use of the coplanarity conjecture. The wavelength can be determined independently to propagation direction only with four closely spaced spacecraft (with separation much smaller than that of Cluster).

An additional wave mode candidate for the high-$\beta$ plasma is a Weibel mode. Pokhotelov and Balikhin (2012) suggested that Weibel mode grows in the finite magnetic field as a mix of two opposite circular polarizations. This variant may be consistent

with the observed irregular variations.

## 5   Conclusions

High-$\beta$ ($\beta > 10$) shocks are relatively rare and largely unexplored class of Earth bow shock. Formation of high-$\beta$ interplanetary plasmas is mostly related with dense slow solar wind and very low magnetic field up to 1–2 nT. The higher is $\beta$ (in OMNI), it is more difficult to confirm it locally. Our shock analysis was limited to oblique and quasi-perpendicular cases and shows some

differences from supercritical shocks with lower $\beta$ both in the general structure of the shock transition and in the properties of magnetic variations. Magnetic field and ion density jumps are smeared to a couple tens of seconds. Dominating magnetic waves have frequencies 0.2–0.5, sometimes, $\sim$2 Hz, irregular, close to linear, polarization and spatial scales around a hundred km. Recent Magnetospheric Multiscale mission observations with very closely spaced spacecraft are necessary to conclude more definitely on the wave mode.

*Author contributions.* OMC and PIS performed the data processing and analysis. AAP is responsible for data analysis and interpretation. AAP prepared the manuscript with contributions from all co-authors.

*Competing interests.* The authors declare that they have no conflict of interest.

*Acknowledgements.* The data analysis was funded with Russian Science Fund project 05-14-00824. We are thankful for Cluster Science Archive, CDAWeb and OMNI for availability of spacecraft data.




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
