# Peer review of "Low frequency magnetic variations at the high- $\beta$ Earth bow shock"

_Annales Geophysicae, 2019_

## Referee Comment (RC1) · Anonymous Referee #1 · 26 Feb 2019

The previous version of the paper did not reach any substantial conclusions. This version of the paper does not reach any substantial conclusions either. It seems that the authors do not realize that simple description of observations is not physics. It is necessary to extract new knowledge from these observations. I inclined to reject the paper. Yet, I would like to give one more chance. Please make sure to show how this study advances our understanding of shock physics beyond just presenting figures and stating that "Magnetic field and ion density jumps are smeared to a couple tens of seconds". Claiming that "High-$\beta$ ($\beta$ > 10) shocks are relatively rare and largely unexplored class of Earth bow shock" is inappropriate for Conclusions. What new physics is found ?

---

## Referee Comment (RC2) · Anonymous Referee #2 · 19 Mar 2019

The paper is about magnetic fluctuations in the Earth's high-beta bow shock. The authors combine data from the Cluster spacecraft and the OMNI database. The topic of high-beta shocks is highly interesting because of their application to astrophysical shocks.

The language in the paper is mostly fluent and clear. The data analysis is detailed and largely precise. However, the presentation of the data is at times lacking and the analysis is sometimes incomplete.

The conclusions mentioned in the paper are that the shock ramp are smeared to 10s of seconds due to the magnetic fluctuations and a somewhat quantitative description of the wave properties of the fluctuations.

[Figure]

I am inclined to consider these conclusions not sufficient to advance our understanding of high-beta shocks and how the fluctuations are different from those in other shocks. The authors should strive to at least qualitatively describe how the high-beta shock fluctuations differ from "normal" shocks and attempt to measure the spatial extent of the fluctuations. The authors should also more carefully try to validate the OMNI values of beta with the Cluster measurements.

Detailed comments followed below:

Introduction, last paragraph:

Page 2, Line 30: The authors should specify how the Geotail, THEMIS, and Interball data were used in this study. This is as far as I can see the last mention of these missions.

This paragraph appears out of place. Such a detailed description of the data used is probably better in the next section.

Page 3, Line ∼3: Motivate why Cluster data was not used directly to calculate beta. Specifically why CIS-HIA/CODIF was not used for density/temperature of the solar wind.

Page 3, Line 5: The authors should clarify what the constant values in OMNI are and how that might affect the validity of the reported beta value.

Page 3, Line 24: The figure referred to is Fig 4.

Page 3, Line 25: "The scatter is indeed large." Include a discussion about why and what it means for the results of the study. Also specify which data were used when both were available. ACE or WIND?

Figure 6: - Panel b: Plots of magnitudes should contain the value 0. - Panel c: The authors should explain why the temperature appears to fall over the shock when the primary role of a shock is to heat the plasma. Also explain why the parallel temperature

is larger than perpendicular in the shock foot. - Panel e: Something seems to be wrong with the ion energy here (even more clear in the supplement). The solar wind speed: 250 km/s -> proton energy: 330 eV -> should be on line between 10ˆ2 and 10ˆ3 which is ~320 eV. But the solar wind is even below 10ˆ2 eV. This must be wrong.

Page 7, Line ~5: I think there is a missed opportunity to determine the shock speed here. It could put a number on shock extent in space, and not just time. There are several ways to calculate shock speed with one spacecraft, see the ISSI book "Analysis Methods for Multi-Spacecraft Data" chapter 10. Also, Cluster 2 encounters the same shock ~1min later. This could be used in a timing analysis.

Page 10, Last paragraph: I think the statement "minimum variance (nominal propagation) direction is well defined" is a bit strong considering the eigenvalue ratio is as big as 0.5.

Page 10, Last paragraph: "The time shift between the magnetic measurements along the maximum variance component is 0.22" It seems that the two time series (Fig 10b) are quite different. The authors should discuss how this might affect the certainty of the timing.

Figure 13: Legends for C3, C4 are missing.

Page 19, Line 13: "The ramp is often used to determine the shock motion with multi-point measurements, but in our case it is impossible". This is not true. With only C3 and C4 it is impossible but in at least one case there is another Cluster satellite that could potentially be used.

Page 19, Line 27: Thus observed variations are strongly different from that in low-beta supercritical events ..." This appears to me to be a strong conclusion. Why is this not mentioned in "Conclusions"?

Page 19, Line 29: "Observed polarization is also not consistent with the earlier suggested alfven mode" also appears to be a strong conclusion. Also it is: "Alfvén".

Page 20, Line 21: "Magnetic field and ion density jumps are smeared to a couple tens of seconds" This is a very imprecise statement. It is completely possible for even low-beta quasi-perp shocks to be so slow that the ramp is smeared to minutes. An attempt to measure the shock ramp in kilometers instead of seconds would be desirable.

Figure S1: - Caption says C4. C4 does not have a functioning CIS-HIA. It should say C3? - Panel b: The OMNI density appears to not match the CIS-HIA density. The authors should try to validate the density with some other instrument, like CIS-CODIF or WHISPER. - Same comment about ion energy in spectrograms.

Figure S3: Same comments as Fig S1.

There are also some minor language errors that I have not listed. Watch out for missing or superfluous "the" as well as the difference between "its" and "it's".

––––––––––––––––––––––––

---

## Author Comment (AC1) · 8 May 2019

Reply to Referee 1

We are grateful to the Referee for attention and inspiring us to make a deeper insight in plasma physics.

Comment

The previous version of the paper did not reach any substantial conclusions. This version of the paper does not reach any substantial conclusions either. It seems that the authors do not realize that simple description of observations is not physics. It is necessary to extract new knowledge from these observations. I inclined to reject the paper. Yet, I would like to give one more chance. Please make sure to show
how this study advances our understanding of shock physics beyond just presenting figures and stating that "Magnetic field and ion density jumps are smeared to a couple tens of seconds". Claiming that "High-$\beta$ ($\beta > 10$) shocks are relatively rare and largely unexplored class of Earth bow shock" is inappropriate for Conclusions. What new physics is found ?

Reply

Perhaps, the referee is not satisfied with the style of Conclusions section. We are aware of three papers addressing specifically high-beta shocks (Formisano et al, 1975, Winterhalter et al, 1988, Farris et al, 1992). Our new quantitative results are actually fully listed in subsection 3.4.

1. First of all, the study systematically addresses the issue of finding high-beta shock examples and proving their beta values. The overall number of potential candidates is estimated. Statistics of Cluster multipoint shock events with small separation is assembled.

2. With 22 Cluster examples, a typical appearance of oblique high-beta shock was well identified, including ramp structure and general properties of variations.

3. Polarization, typical frequency, in some cases, the spatial scale of dominating magnetic variations was determined. Two distinct variants of wave structure were found, with dominating 0.5 Hz and 2 Hz waves. (Note, that the only wave property reported before was "very high amplitude").

4. Comparison with the previous results shows distinct difference of observed variations from that previously reported in low beta shocks, in particular, it is not consistent with fast magnetosonic or Alfven waves. There is some consistency with Weibel mode.

What was really not done, is some "ultimate identification of wavemode", requiring reliable wave vector determination. It, of course, remains one of our primary goals in the following studies. Perhaps this was meant by the Referee as absence of new

physics. We, however, can not expect that all problems of high-beta shocks will be solved in the first paper. Hundreds of papers were written on more ordinary shocks, and still many things are not clear. We make the conclusions more specific and also enhance interpretations (though more speculative at this stage) in Discussion.

Changes in the text

(In Discussion):

. . . A shock example #3 with very low upstream magnetic field about 1 nT exhibits very variable direction of magnetic field both upstream and downstream, complicating definition of shock magnetic geometry. This issue might be of a special interest, since at some (very small) value of magnetic field, it's direction should become unimportant for the shock structure. Then, there will be no difference between perpendicular and parallel shocks and the shock spatial scale, as well as ion escape upstream, should be defined with some nonmagnetic parameters. Our study suggests preliminarily, that transition to 'non-magnetic' case may occur as in Example #3, at $\beta$ of several tens. Thus, one of the main topics for the future studies is to observe in greater detail the dependence of the shock scales for the events with the extreme large $\beta$....

. . .An alternative wave mode candidate, frequently suggested for high-$\beta$ plasma, is Weibel instability. With no seed magnetic field, the Weibel mode has only imaginary frequency. The latter means that the magnetic field clamps are growing faster than propagate. For finite magnetic field **?** suggested, that Weibel mode grows as a mix of two opposite circular polarizations, attains some small real part of frequency and is fundamentally similar to drift mirror mode. Thus, in some features (linear polarization, chaotic phase) it is consistent with observations.

(In conclusions): High-$\beta$ ($\beta > 10$) shocks are relatively rare and largely unexplored class of Earth bow shock. Formation of high-$\beta$ interplanetary plasmas is mostly related with dense slow solar wind and very low magnetic field up to 1–2 nT. Due to spatial variability of low IMF, it is more difficult to determine shock geometry for higher $\beta$ (in
OMNI) cases. However, at some (large) $\beta$ shock structure should become independent from magnetic field direction. This is an interesting direction of future studies.

In oblique and quasi-perpendicular shocks, the main magnetic field and ion density jumps are smeared to several hundred km. Dominating magnetic variations have amplitudes much larger than the background field, frequencies 0.2–0.5 Hz, sometimes, $\sim$2 Hz. Polarization is mostly irregular and close to linear. The spatial scales range from several tens to couple hundred km. These properties are definitely different from that for fast magnetosonic or Alfvén modes earlier reported for other shock types. In some features the variations may be consistent with the Weibel instability, but observations with more closely spaced spacecraft are necessary to conclude more definitely on the wave mode.

---

## Author Comment (AC2) · 8 May 2019

**Replies to Referee #2.**

We would like to thank the Referee for the attentive reading and several useful suggestions.

*Comment*

*The paper is about magnetic fluctuations in the Earth's high-beta bow shock. The authors combine data from the Cluster spacecraft and the OMNI database. The topic of high-beta shocks is highly interesting because of their application to astrophysical shocks. The language in the paper is mostly fluent and clear. The data analysis is detailed and largely precise. However, the presentation of the data is at times lacking and the analysis is sometimes incomplete. The conclusions mentioned in the paper are that the shock ramp are smeared to 10s of seconds due to the magnetic fluctuations and a somewhat quantitative description of the wave properties of the fluctuations.*

*I am inclined to consider these conclusions not sufficient to advance our understanding of high-beta shocks and how the fluctuations are different from those in other shocks. The authors should strive to at least qualitatively describe how the high-beta shock fluctuations differ from "normal" shocks and attempt to measure the spatial extent of the fluctuations. The authors should also more carefully try to validate the OMNI values of beta with the Cluster measurements.*

**Reply**

We should first mention, that up to now observational information on high-beta shocks was very limited (mostly three papers: Formisano et al, 1975, Winterhalter  et al, 1988, Farris et al, 1992). The wave analysis was not performed at all. These papers only checked some very general shock parameters and noted presence of high-amplitude turbulence (in comparison with lower-beta shocks). We do not count here several reports addressing beta=2 shocks as high-beta ones (like Scudder et al, 1986). Therefore, we consider, that our study in fact makes a significant advance for high-beta shocks:

1. First of all, the study systematically addresses the issue of finding high-beta shock examples and proving their beta values. The overall number of potential candidates is estimated. Statistics of Cluster multipoint shock events with small separation is assembled.

2. With 22 Cluster examples, a typical appearance of oblique high-beta shock was well identified, including ramp structure and general properties of variations.

3. Polarization, typical frequency, in some cases, the spatial scale of dominating magnetic variations was determined. Two distinct variants of wave structure were found, with dominating 0.5 Hz and 2 Hz waves. (Note, that the only wave property reported before was "very high amplitude").

4. Comparison with the previous results shows distinct difference of observed variations from that previously reported in low beta shocks, in particular, it is not consistent with fast magnetosonic or Alfven waves. There is some consistency with Weibel mode.

What was really not done, is some "ultimate identification of wavemode", requiring reliable wave vector determination. It, of course, remains one of our primary goals in the following studies.

We intentionally do not directly compare with lower beta shocks examples.  We compare just with previous publications. This is done for two reasons:

1. We are not interested so much in finding **the differences** with lower beta shocks. The comparison was done only to highlight specific properties of our variations. We are interested in determining properties of high-beta shocks as such, having in mind their cross-discipline importance. In fact, the larger is beta, more interesting is the study.

2. There exist thousands of crossings of low-beta shocks with very diverse properties. References in our report partially illustrate this diversity of conclusions about fluctuation modes and ramp structure. The only well bounded classes are supercritical q-perp shocks with domination of whistler mode fluctuations and q-par shocks. Beyond these two classes, it is totally unclear which low-beta shocks to choose for comparison. Proper selection of low beta examples would be more laborious, than this study of high-beta events. Thus we consider, that inclusion of additional low-beta examples will strongly deviate us from the main goal.

Issue of validation of OMNI beta values is addressed below among more specific comments. We also added estimates of shock speed, spatial length of ramp and WHISPER density. We also somewhat reworded Discussion and Conclusions to make them more specific, to include more interpretation specific to high-beta shocks.

**Changes in the text:**

(In Discussion):

… A shock example \#3 with very low upstream magnetic field about 1 \unit{nT} exhibits very variable direction of magnetic field both upstream and downstream, complicating definition of shock magnetic geometry. This issue might be of a special interest, since at some (very small) value of magnetic field, it's direction should become unimportant for the shock structure. Then, there will be no difference between perpendicular and parallel shocks and the shock spatial scale, as well as ion escape upstream, should be defined with some nonmagnetic parameters. Our study suggests preliminarily, that transition to 'non-magnetic' case may occur as in Example \#3, at $\beta$ of several tens. Thus, one of the main topics for the future studies is to observe in greater detail the dependence of the shock scales  for the events with the extreme large $\beta$....

…An alternative wave mode candidate, frequently suggested for high-$\beta$ plasma, is Weibel instability. With no seed magnetic field, the Weibel mode has only imaginary frequency. The latter means that the magnetic field clamps are growing faster than propagate. For finite magnetic field \citet{pokhotelov2012} suggested, that Weibel mode grows as a mix of two opposite circular polarizations, attains some small real part of frequency and is fundamentally similar to drift mirror mode. Thus, in some features (linear polarization, chaotic phase) it is consistent with observations.

(In conclusions): High-$\beta$ ($\beta>10$) shocks are relatively rare and largely unexplored class of Earth bow shock. Formation of high-$\beta$ interplanetary plasmas is mostly related with dense slow solar wind and very low magnetic field up to 1--2 \unit{nT}. Due to spatial variability of low IMF, it is more difficult to determine shock geometry for higher $\beta$ (in OMNI) cases. However, at some (large) $\beta$ shock structure should become independent from magnetic field direction. This is an interesting direction of future studies.

In oblique and quasi-perpendicular shocks, the main magnetic field and ion density jumps are smeared to several hundred \unit{km}. Dominating magnetic variations have amplitudes much larger than the background field, frequencies 0.2--0.5 \unit{Hz}, sometimes, $\sim$2 \unit{Hz}. Polarization is mostly irregular and close to linear. The spatial scales range from several tens to couple hundred \unit{km}. These properties are definitely different from that for fast magnetosonic or Alfv\'{e}n modes earlier reported for other shock types. In some features the variations may be consistent with the Weibel instability, but observations with more closely spaced spacecraft are necessary to conclude more definitely on the wave mode.

***\*\*\*\****

***Comment:***

*Page 2, Line 30: The authors should specify how the Geotail, THEMIS, and Interball data were used in this study. This is as far as I can see the last mention of these missions. This paragraph appears out of place. Such a detailed description of the data used is probably better in the next section.*

**Reply and change in the text:**

OK. Data description paragrpahs were moved to Section 2. Mention of "Geotail, THEMIS, and Interball" is deleted, since it appears to be confusing. These spacecraft were used to understand the shock statistics during the whole analyzed period  1995-2017, but in this study only Cluster multipoint examples are presented.

***\*\*\*\****

*Comment:*

*Page 3, Line ~3: Motivate why Cluster data was not used directly to calculate beta. Specifically why CIS-HIA/CODIF was not used for density/temperature of the solar wind.*

**Reply:**

HIA/CODIF data were not used to calculate local beta for two reasons. (1) Presence of even a small number of reflected ions strongly changes upstream temperature. (2) OMNI ion data are provided by dedicated solar wind instruments (retarding analyzers), while HIA/CODIF are general-purpose deflecting spectrometers. The latter are not suited to measure reliably tiny solar wind ion temperature. Considering validity of Cluster solar wind density and temperature goes far beyond purpose of this paper. Since in the solar wind magnetic field is more variable than density, we consider it is more reliable to use verified OMNI beta data and recheck comparing Cluster and OMNI magnetic field as explained in the paper. However we now included WHISPER density data, where available, for illustration.

**Change in the text:**

AS an additional illustration we now show WHISPER electron density data in solar wind. It is almost identical to OMNI density. Updated figures 6, S1, S3 are added in the end of this text.

***\*\*\*\****

*Comment:*

*Page 3, Line 5: The authors should clarify what the constant values in OMNI are and how that might affect the validity of the reported beta value.*

**Reply:**

Added. These values are multiyear averages, deviation from them may slightly modify actual beta. However, at this stage we left this issue (OMNI beta values) as is. Among other problems with beta, this source of errors is not the primary one.

**Change in the text:**

(in Sec.2): $\beta$ values are precalculated in OMNI-2, assuming constant electron temperature (140000 K), He++ fraction (0.05) and He++ temperature (four times larger than proton temperature).

(In Sec.4): Assumptions on the constant helium content and constant electron temperature, used while OMNI $\beta$ calculations may also result in some errors. For example, a factor of two change in electron temperature will result in the change of $\beta$ by about 30\%.  Factor of two variations of the He++ content will result in variations of $\beta$ around 10\%.

***\*

***Comment:***

*Page 3, Line 24: The figure referred to is Fig 4.*

**Reply:**

Thanks, corrected

***\*

***Comment:***

*Page 3, Line 25: "The scatter is indeed large." Include a discussion about why and what it means for the results of the study. Also specify which data were used when both were available. ACE or WIND?*

**Reply:**

OMNI data were always used. We improved description of statistics. Right in the next paragraph it is stated (*what it means for the results?*), that local beta confirmation is always necessary. Full discussion is already included in the relevant section.

**Change in the text:**

Fig. 4 shows comparison of $\beta$ calculation for Wind and ACE 1-hour data (only for times, when Wind data were used in OMNI).For this OMNI-2 subset there were 618 1-hour points, with $\beta>$10 either in Wind or ACE data. Only for 196 of them difference between $\beta$ at two spacecraft was less than 30\%. For more than half of events (318) difference between the spacecraft was larger than 50\%. Such substantial difference is partially due to quadratic dependence of $\beta$ on magnetic field.

***\*

***Comment:***

*Figure 6: - Panel b: Plots of magnitudes should contain the value 0.*

**Reply:**

This depends on the scope of change in the variable. However, we rechecked all panels and included "zero" everywhere, where reasonable. Updated Figures 6, S1, S3 are provided in the end of this text.

***\*

***Comment:***

*- Panel c: The authors should explain why the temperature appears to fall over the shock when the primary role of a shock is to heat the plasma. Also explain why the parallel temperature is larger than perpendicular in the shock foot.*

**Reply:**

We changed the scale of temperature plot to logarithmic one. Perpendicular temperature steadily grows across the transition. Parallel temperature maximizes right upstream the shock, because of the presence of upstream-moving ions (probably mostly field-aligned). The spread of velocity between solar wind flow and these ions creates a false increase of ion parallel temperature.

**Change in the text:**

The scale of temperature panel is changed to log. (See updated figures in the end of this text).

(in Text): Cluster ion (proton) density in solar wind is lower, than that in OMNI, however WHISPER electron density is almost the same. Proton perpendicular temperature grows as expected towards downstream, while the parallel temperature peaks just upstream the shock front. We attribute this peak to the upstream-moving field-aligned protons. The presence of two populations in the distribution function with strongly different flow velocity, results in the false temperature increase.
* * *
**Comment:**

*-Panel e: Something seems to be wrong with the ion energy here (even more clear in the supplement). The solar wind speed: 250 km/s -> proton energy: 330 eV -> should be on line between 10^2 and 10^3 which is ~320 eV. But the solar wind is even below 10^2 eV. This must be wrong.*

**Reply:**

Ok. Thanks a lot. Energy values were twice subjected to logarithm by mistake. Now corrected. Updated figures are attached in the end of this text.
* * *
**Comment:**

*Page 7, Line ~5: I think there is a missed opportunity to determine the shock speed here. It could put a number on shock extent in space, and not just time. There are several ways to calculate shock speed with one spacecraft, see the ISSI book " Analysis Methods for Multi-Spacecraft Data" chapter 10. Also, Cluster 2 encounters the same shock~1min later. This could be used in a timing analysis.*

**Reply:**

OK, thanks for the idea! Indeed, this will provide some additional information. Cluster 2 crosses the shock within two minutes. However, such determination of shock speed is not very reliable quantitatively. Nonetheless we include this information in the text.

As concerns single spacecraft methods, they returned very differing results for the shock speed around 100 km/s. In fact, it is not surprising. These formulas manipulate with velocity of the order of hundreds km/s to get the result of the order of ten km/s. With noisy data, computations of such kind are prone to errors and seldom used. Thus we choose not to include these estimates.

**Change in the text:**

(Event1): However, Cluster 2, being about 6000 km away from the pair C3 and C4, crossed the shock two minutes later (exact values are 6231 km and 124 s between C3 and C2, not shown here). The separation along the model normal is 1032 km, the spacecraft are placed almost along the shock front. Shock velocity along the normal is then 8.3 km/s outbound. This calculation is not very reliable for two reasons: (1) The spacecraft are mostly separated along the front by about 6000 km, shock motion may be different in two so different points. (2) The reverse crossings occurred less than 10 min later, thus shock speed might substantially change on a scale of two min (separation between C3 and C2). Nevertheless, one can estimate the spatial scale of the smeared ramp. Duration of 35 s corresponds to 290 km. The gyroradius of solar wind proton is $\sim$700 km, proton inertial scale is $\sim$80 km.

(Event2): Similar to Example 1, one can estimate shock speed along the normal, comparing with C2 crossings (not shown here). However for this example, the spacecraft separation is more than 11000 km, while the separation along the model normal is much smaller, just about 100 km. The estimated shock speed is 1.5--2.2 km/s (comparing the pairs C3-C3 and C4-C2), corresponding to the smeared ramp

width about 50 km. However this estimate is very unreliable, since the calculation result would strongly depend on small variations of the actual normal.

(Event3): Similar to Example 1, one can estimate shock speed along the normal, comparing with C2 crossing (not shown here).  The spacecraft separation is 5700--5800 km, while the separation along the model normal is about 1400 km. The estimated shock speed is about 11 km/s, corresponding to the smeared ramp width about 330 km. The gyroradius  of solar wind proton is about 2000 km, due to very low IMF, proton inertial scale is $\sim$80 km.

(Discussion): For example the main magnetic and ion density increase in the shock front was for all observed event around always several tens of seconds or several hundred km for typical shock proper velocity of about 10 km/s. This scale is larger, than proton inertial length, but smaller than ion gyroradius in solar wind.  Dependence of shock spatial scale on $\beta$ is an interesting moment and should be addressed in future studies on larger statistics.
* * *
*Comment:*

*Page 10, Last paragraph: I think the statement "minimum variance (nominal propagation) direction is well defined" is a bit strong considering the eigenvalue ratio is as big as 0.5.*

**Reply:**

Here the intermediate-to-maximum ratio is 0.5, while the minimum-to-intermediate ratio, defining quality of minvar, is small enough - 0.15. So, the statement is correct.
* * *
*Comment:*

*Page 10, Last paragraph: "The time shift between the magnetic measurements along the maximum variance component is 0.22" It seems that the two time series (Fig 10b) are quite different. The authors should discuss how this might affect the certainty of the timing.*

**Reply:**

We apply the correlation analysis on a small interval 160115-160125 only, since indeed on other intervals the differences are large. On the given interval the large-scale lower-frequency structures are similar, while the most of differences are due to higher-frequency and lower-amplitude variations. In fact this issue is mentioned in the very beginning of description, since this example was chosen to show a case with substantial difference of magnetic variations.

**Change in the text:**

(In Section 3.2): The specific feature of this event is a difference of C3 and C4 oscillations during the first 20 \unit{s} downstream the front (16:01:25--16:01:35 UT), despite relatively small separation. The presence of such difference in amplitudes is typical for all shocks registered during this day (8 crossings within 2 hours in Table S1). Despite the differences, it is possible to perform multipoint separation analysis for the interval 16:01:15-16:01:25, where two waveforms in $B_y$ component (Fig. 10b) are kind of similar and shifted by a fraction of period.

(In Section 4.3): The second variant of a spatial scale is illustrated with Example 2. It includes the mix of scales of the order of hundred \unit{km}, which can be captured with our spacecraft separation, and of the order of tens \unit{km}. As a result, the waveforms are rather different, but common features can sometimes be traced. Finally the third variant (Example 3) suggests the dominating spatial scale of at most 200 \unit{km}.
* * *
***Comment:***

*Figure 13: Legends for C3, C4 are missing.*

**Reply and change in the text:** OK, corrected.
* * *
***Comment:***

*Page 19, Line 13: "The ramp is often used to determine the shock motion with multipoint measurements, but in our case it is impossible". This is not true. With only C3 and C4 it is impossible but in at least one case there is another Cluster satellite that could potentially be used.*

**Reply:**

Determination of shock velocity is commented above. Here, we would like to insist, that the ramp can not be used to determine the scale with C3-C4 timing. With C2-C3 or C2-C4 timing the detailed form of ramp is unimportant, since the time difference is large (2 min), so even completely different form of the main jump (within couple of seconds) will not affect the computational result.
* * *
***Comment:***

*Page 19, Line 27: Thus observed variations are strongly different from that in low-beta supercritical events ..." This appears to me to be a strong conclusion. Why is this not mentioned in "Conclusions"?*

*Page 19, Line 29: "Observed polarization is also not consistent with the earlier suggested alfven mode" also appears to be a strong conclusion. Also it is: "Alfvén".*

**Reply:**

See general comment in the beginning. Conclusions are now reformulated. We state, that the observed variations are not alike those previously reported for low-beta shocks.
* * *
***Comment:***

*Page 20, Line 21: "Magnetic field and ion density jumps are smeared to a couple tens of seconds" This is a very imprecise statement. It is completely possible for even low beta quasi-perp shocks to be so slow that the ramp is smeared to minutes. An attempt to measure the shock ramp in kilometers instead of seconds would be desirable.*

**Reply:**

Done. See reply above.
* * *
***Comment:***

*Figure S1: - Caption says C4. C4 does not have a functioning CIS-HIA. It should say C3? - Panel b: The OMNI density appears to not match the CIS-HIA density. The authors should try to validate the density with some other instrument, like CIS-CODIF or WHISPER. - Same comment about ion energy in spectrograms.*

*Figure S3: Same comments as Fig S1.*

**Reply:**

OK. This is a misprint in captions of S1 and S3. Example 1 uses C4 CODIF, Example 2 and 3 – C3 HIA as written on spectrograms.

Solar wind density is not a strong side of Cluster measurements. We rechecked with WHISPER electron density, it is closer to OMNI, than ion density, however we do not use local density for any quantitative conclusions.

**Text change:**

Figure captions for S1 and S2 are corrected. WHISPER density is included in Figures for two examples, where it was available.  A short notice in the text is also included. New Figures 6, S1 and S3 are shown in the end of this file.
* * *
*Comment:*

*There are also some minor language errors that I have not listed. Watch out for missing or superfluous "the" as well as the difference between "its" and "it's".*

**Reply:**

Done as we can find.

**New figures**

[Figure]

Figure 6. Overview of C4 magnetic and plasma (CODIF) measurements for event 18 December 2011. (a) proton velocity, (b) proton density, OMNI solar wind density, WHISPER electron density, (c) proton parallel and perpendicular temperature, (d) magnetic field magnitude and OMNI IMF magnitude, (e,f) proton spectrograms for the sunward and dawnward looking sectors.

[Figure]

Fig. S1. Overview of C3 magnetic and plasma (HIA) measurements for event 04 January 2008. (a) ion velocity, (b) ion density and OMNI solar wind density, (c) ion parallel and perpendicular temperature, (d) magnetic field magnitude and OMNI IMF magnitude, (e,f) ion spectrograms for the sunward and dawnward looking sectors.

[Figure]

Fig. S3. Overview of C3 magnetic and plasma (HIA) measurements for event 03 January 2008. (a) ion velocity, (b) ion density and OMNI solar wind density, (c) ion parallel and perpendicular temperature, (d) magnetic field magnitude and OMNI IMF magnitude, (e,f) ion spectrograms for the sunward and dawnward looking sectors.

---

## Referee Report (RR1)

**Report: angeo-2019-7**

It seems that the authors do not understand that simple description of the observations is not new physics yet. I give up on that. It is possible that others will be able to use the data to arrive at new physics. Yet, even primary data analysis should be done carefully and not as it is now.

Provide the full list of high-beta shocks as a supplement material.

Event 1:

- "shock velocity is definitely much higher than the spacecraft velocity" - relative to what ?

- "Fig. 6 contains overview of magnetic field and plasma parameters." - does not seem that quiet downstream is reached.

- Table S1: X,Y,Z are not relevant to the analysis. The following parameters which ARE relevant are absent: Alfven speed, shock speed relative to the upstream plasma, Bd/Bu, nd/nu, thermal speed (replace T), ion inertial length, ion thermal gyroradius, ion convective gyroradius.

- "The coplanarity calculation for the shock normal" - provide information about the region used for coplanarity.

- Provide estimates of errors of theta and Mach number determination

-"The final value of downstream magnetic field is around 10 nT" - where is this "final downstream" ?

-"first signs of gyrating ions upstream" - reflected-gyrating ions can be observed at distances of the order of the ion convective gyroradius upstream of the ramp (see below), gyrophase bunched beams propagate toward upstream. What gyrating ions are mentioned in the paper ? Bale et al, PRL 91, 265004, 2003, doi:10.1103/PhysRevLett.91.265004 show that the density transition width is of the order of the ion convective gyroradius. Why special is found for this shock ?

- "The increase in magnetic field magnitude (aka shock ramp in a quasi-perpendicular case) is smeared" - magnetic ramp is defined as the region of the largest magnetic increase. It is between 14:37:48-14:37:49 in Figure 7 and not "smeared". The mentioned half a minute is the precursor region including foot and shock generated magnetic fluctuations. See Scudder et al, 1986.

- "Shock velocity along the normal is 8.3 km/s outbound" - how this is calculated ?

- "This calculation is not very reliable" - yet it is used for estimates of spatial scales. What are the errors ?

-"Despite the described smeared magnetic field increase, the full shock transition is rather compact and coherent and thus it is distinctly different from what expected for quasi-parallel shock with multiple shocklets" - provide quantitative description: what is "smeared", what is "compact" and "coherent" ? It seems that this sentence contradicts earlier statements.

- The frequency spectra in Figure 8 seem to be self-similar power-law spectra with a low frequency cutoff. This is typical for a broad-band Fourier transform of the large-amplitude magnetic fluctuations. The "dominant frequency" is just the inverse of the peak-to-peak time (about 2 s visually).

- "polarization actually might be linear with the variable eigenvector" - what is this ? Please split the region of calculation to sub-regions to support the statement.

- "We also estimate the span of principally possible wavelengths." - should be directly compared to physically meaningful spatial scales.

Event 2: same comments as above. Additional comments:

- "where two waveforms in By component (Fig. 10b) are kind of similar and shifted by a fraction of period." - they do not look similar at all. Actually, measurements of the two spacecraft look quite different. If you still think they can be considered as similar but shifted please show these two profiles overplotted when properly shifted.

- "shows absence of any stable polarization, which can be in- terpreted as sometimes linear, sometimes circular" - usually identification of polarization is done using several wavelength. What is "sometimes" ?

Event 3: same comments as above. In addition, the magnetic field looks quite different from the previous two shocks and now clear asymptotic downstream is seen. What are the errors in determination of theta ? Can this shock be a quasi-parallel and/or strongly nonstationary ?

Conclusions: "shock structure should become independent from magnetic field direction" - how did you arrive at this conclusion ?

The above comments are also addressed to Discussion and Conclusions where many of the earlier statements are simply reiterated. The paper also claims that "overall layout is quite characteristic and distinctly different from that for super-critical quasi-perpendicular shocks." It is not explained what is "characteristic" (see e.g. Scudder et al, 1986 as an example of how a "typical" shock should be described). "Distinct difference" does not seem to be supported by observations.

English is poor, please edit.

---

## Referee Report (RR2)

**Referee report: Low frequency magnetic variations at high-β Earth bow shocks**

The authors have carefully responded to all of my previous comments in a satisfying way. There has been a clear improvement in the presentation of the conclusions, validating the solar wind beta values, and determining spatial scales.

The manuscript still has some problems with language and presentation which makes it difficult to read. In particular, the manuscript contains several language errors (not least the abstract) and require a thorough proof-reading.

I recommend the paper for publication after these problems are resolved.

See more detailed comments below:

Title: The authors should consider referring to the "magnetic variations" as "magnetic fluctuations" instead. I think the word "variation" is better suited for large-scale or solar wind driven changes. I also think the term "bow shocks" is somewhat strange and confusing. It's all the same bow shock even if there are several crossings, so the plural form is unnecessary. Maybe "... at the Earth's high-beta bow shock" reads better?

Page 1, Line 2: The Earth's bow shock is not ubiquitous in the universe. Better to say "However such shocks are ubiquitous..."

Page 1, Line 3: "IMF" is an abbreviation and should be explained if used.

Page 1, Line 4-5: "About a hundred high-beta shocks were initially identified during 1995–2016...": Is this initial list crucial to the results of the manuscript? If not, then it should not be in the abstract. The information here is also not consistent with that stated in the paper (2016 or 2017?).

Page 1, Line 5-6: Multi-point observations are presented for three shock crossings by Cluster, not 22. I think the information about these three events which are more closely studied is much more crucial than the (even more initial) list of ~100 crossings.

Page 1, Line 9: "Their polarization has...": This should refer to the sentence before which does not work. Rather say "The wave polarization...".

Page 1, Line 10-11: "Spatial scales...": It should be made clear if this refers to the extent where the fluctuations are found or the scales (wavelengths) of the waves. This information is also repeated in the abstract.

Page 2, Line 30: First mention of "IMF" in text. Should be explained.

Page 4, Line 7: "We scanned 1995–2017 observations by all available spacecraft": The authors removed the mention of other spacecraft (THEMIS, etc), which makes the years 1995-2017 confusing here, since Cluster was launched in 2000. I understand that the final list of 22 events is a selection from the first list of about a hundred events (an exact number would be better). However, the authors must clarify which spacecraft made up the initial list if it is mentioned.

Page 5, Line 15-20: I think there should be a short explanation that HIA/CODIF data in the solar wind may be unreliable and why the OMNI beta-value was rather used. Additionally, now WHISPER data is also used in the manuscript. That instrument should be mentioned and referenced here as well.

Page 5, Line 18: The CIS data guide states that the time resolution of CIS-HIA and CIS-CODIF is 4 s. Can you double-check the time-resolution of the data in your events and rephrase here if needed?

Page 22, Line 32: I'm guessing it should read "low-beta" instead of "supercritical" as all the shocks studied here are also supercritical.

Several more language issues that are not listed here.

---

## Author Response (AR2)

We thank both Referees and Editor for useful suggestions and kind help with the language. Only principal changes are marked by bold. Figures 2,6,9, S1,S3 are updated.

**Comments of Ref. 2**

*Title: The authors should consider referring to the "magnetic variations" as "magnetic fluctuations" instead. I think the word "variation" is better suited for large-scale or solar wind driven changes. I also think the term "bow shocks" is somewhat strange and confusing. It's all the same bow shock even if there are several crossings, so the plural form is unnecessary. Maybe "... at the Earth's high-beta bow shock" reads better?*
*Page 1, Line 2: The Earth's bow shock is not ubiquitous in the universe. Better to say "However such shocks are ubiquitous..."*
*Page 1, Line 3: "IMF" is an abbreviation and should be explained if used.*

Done

*Page 1, Line 4-5: "About a hundred high-beta shocks were initially identified during 1995–2016...": Is this initial list crucial to the results of the manuscript? If not, then it should not be in the abstract. The information here is also not consistent with that stated in the paper (2016 or 2017?).*

Deleted from abstract

*Page 1, Line 5-6: Multi-point observations are presented for three shock crossings by Cluster, not 22. I think the information about these three events which are more closely studied is much more crucial than the (even more initial) list of ~100 crossings.*

Corrected to: In this report three characteristic crossings by Cluster project (out of 22 found). 100 events were mentioned only to give a sense of total possible statistics.

*Page 1, Line 9: "Their polarization has...": This should refer to the sentence before which does not work. Rather say "The wave polarization...".*

Done

*Page 1, Line 10-11: "Spatial scales...": It should be made clear if this refers to the extent where the fluctuations are found or the scales (wavelengths) of the waves. This information is also repeated in the abstract.*

Done. Wavelength, if this term is applicable to a not very periodic signal.

*Page 2, Line 30: First mention of "IMF" in text. Should be explained.*

Done.

*Page 4, Line 7: "We scanned 1995–2017 observations by all available spacecraft": The authors removed the mention of other spacecraft (THEMIS, etc), which makes the years 1995-2017 confusing here, since Cluster was launched in 2000. I understand that the final list of 22 events is a selection from the first list of about a hundred events (an exact number would be better). However, the authors must clarify which spacecraft made up the initial list if it is mentioned.*

Replied above

Added text: "However, all these events still need a more detailed confirmation, in particular, of local high $\beta$, stable enough crossing velocity, plasma data availability etc."

*Page 5, Line 15-20: I think there should be a short explanation that HIA/CODIF data in the solar wind may be unreliable and why the OMNI beta-value was rather used. Additionally, now WHISPER data is also used in the manuscript. That instrument should be mentioned and referenced here as well.*

WHISPER reference included. Details of comparison of OMNI and local data are available in the Event description section.

*Page 5, Line 18: The CIS data guide states that the time resolution of CIS-HIA and CIS-CODIF is 4 s. Can you double-check the time-resolution of the data in your events and rephrase here if needed?*

Rechecked. Actually it is 4-12 sec, depending on a case and on a parameter (some spectra are 12-sec).

*Page 22, Line 32: I'm guessing it should read "low-beta" instead of "supercritical" as all the shocks studied here are also supercritical.*

Conclusion is reworded.

**Comments of Referee 1.**
*Report: angeo-2019-7 It seems that the authors do not understand that simple description of the observations is not new physics yet. I give up on that. It is possible that others will be able to use the data to arrive at new physics. Yet, even primary data analysis should be done carefully and not as it is now.*
*Provide the full list of high-beta shocks as a supplement material.*

The "full" number of shocks (about 100) is given only as an illustration of abundance. All these candidate events still need to be rechecked for usability. This will be gradually done in the future publications.

*Event 1: - "shock velocity is definitely much higher than the spacecraft velocity" - relative to what ?*

Velocity of shock motion due to solar wind irregularities, etc in the geophysical frame. Now deleted to avoid confusion.

*"Fig. 6 contains overview of magnetic field and plasma parameters." - does not seem that quiet downstream is reached.*
The interval, where downstream parameters were averaged is 14:36-14:37 UT, for upstream – 14:40-14:41 UT.  Here all transition effects in plasma moments are over. Information is added in the text. The figure is replaced to prove, that quiet downstream is reached.

*- Table S1: X,Y,Z are not relevant to the analysis. The following parameters which ARE relevant are absent: Alfven speed, shock speed relative to the upstream plasma, Bd/Bu, nd/nu, thermal speed (replace T), ion inertial length, ion thermal gyroradius, ion convective gyroradius.*

All further parameters can be calculated using given OMNI solar wind data. Mach numbers, lengths and compression ratios are in the text for the particular shock examples in the text, where relevant. The shock hydrodynamics is not our primary goal, so these values were not calculated for all 22 shocks.

*- "The coplanarity calculation for the shock normal" - provide information about the region used for coplanarity.*

Done.

*- Provide estimates of errors of theta and Mach number determination*

It is not clear what kind of errors the Referee is speaking about. Definitely uncertainties due to magnetic field and solar wind variations are larger than the nominal measurement errors. The most critical are that for shock geometry. We estimated them comparing various methods and data sources. The resulting uncertainty in theta is about 10 degrees. This information is available in the text.

*-"The final value of downstream magnetic field is around 10 nT" - where is this "final downstream" ? –*

More detailed information is included in the relevant place.

*"first signs of gyrating ions upstream" - reflected-gyrating ions can be observed at distances of the order of the ion convective gyroradius upstream of the ramp (see below), gyrophase bunched beams propagate toward upstream. What gyrating ions are mentioned in the paper?*

Indeed this statement is somewhat misleading. We now reformulate it as "upstream high-energy ions, which can be observed on the spectrogram in Fig. 6f". We are not interested in this investigation to analyze details of ion kinetics.

*Bale et al, PRL 91, 265004, 2003, doi:10.1103/PhysRevLett.91.265004 show that the density transition width is of the order of the ion convective gyroradius. Why special is found for this shock?*

OK, finally thanks a lot for the useful reference. Comparisons with these scales are added. In fact, observed shocks obey Bale et al statistics. Visually the ramp is smeared because of much larger gyroradius and relatively low shock speed. Note however, that this is a side result for these investigation, we are focused on variations.

*- "The increase in magnetic field magnitude (aka shock ramp in a quasiperpendicular case) is smeared" - magnetic ramp is defined as the region of the largest magnetic increase. It is between 14:37:48-14:37:49 in Figure 7 and not "smeared". The mentioned half a minute is the precursor region including foot and shock generated magnetic fluctuations. See Scudder et al, 1986.*

This comment actually contradicts the previous one. The mentioned half a minute contains not only magnetic field increase, which is indeed wavy and allows diverse interpretations, but also a steady increase of density, typical for ramp, not for foot.

*- "Shock velocity along the normal is 8.3 km/s outbound" - how this is calculated ? - "This calculation is not very reliable" - yet it is used for estimates of spatial scales. What are the errors ?*

Computed as delay between C2 and C3. The source of uncertainties is explained. Basically distance between C2 and C3 is too large to assure constant shock velocity on this scale.

*-"Despite the described smeared magnetic field increase, the full shock transition is rather compact and coherent and thus it is distinctly different from what expected for quasi-parallel shock with multiple shocklets" - provide quantitative description: what is "smeared", what is "compact" and "coherent" ? It seems that this sentence contradicts earlier statements.*

Here we merely state, that the shock is clearly different from the q-parallel variant. Now it is moved to discussion and reformulated to avoid confusion.

*- The frequency spectra in Figure 8 seem to be self-similar power-law spectra with a low frequency cutoff. This is typical for a broad-band Fourier transform of the large-amplitude magnetic fluctuations. The "dominant frequency" is just the inverse of the peak-to-peak time (about 2 s visually).*

We do not see any controversy here. Of course, dominant frequency is inverse peak-to peak time. The main conclusion, is that this peak-to peak time is rather stable (narrow maximum).

*1 - "polarization actually might be linear with the variable eigenvector" - what is this ? Please split the region of calculation to sub-regions to support the statement.*

Reworded as "linear with the maximum variance direction changing every several periods (two variants are shown by red lines)"
The change of the maximum variance direction is clear from the hodograph in Fig. 9, it is additionally marked up with red lines.

 *- "We also estimate the span of principally possible wavelengths." - should be directly compared to physically meaningful spatial scales.*

All local scales are actually computed in the previous paragraphs. Now we repeat values also here. However, since the span is large, interpretation is difficult.

*Event 2: same comments as above.*
*See above*
*Additional comments: - "where two waveforms in By component (Fig. 10b) are kind of similar and shifted by a fraction of period." - they do not look similar at all. Actually, measurements of the two spacecraft look quite different. If you still think they can be considered as similar but shifted please show these two profiles overplotted when properly shifted.*

While indeed for this event waveforms are "not so similar", we still find one small interval (time is given in the text), where calculation is possible, because a small persistent time shift (equal to a fraction of a period) is easily observable.  The alternative is to do nothing.

*- "shows absence of any stable polarization, which can be interpreted as sometimes linear, sometimes circular" - usually identification of polarization is done using several wavelength. What is "sometimes" ?*

We believe it is clear after visual inspection of hodograph in Fig 11.

*Event 3: same comments as above. In addition, the magnetic field looks quite different from the previous two shocks and now clear asymptotic downstream is seen.*
*What are the errors in determination of theta ? Can this shock be a quasi-parallel and/or strongly nonstationary ?*

Errors (better to say, uncertainties) are due to almost 90 deg uncertainty of upstream magnetic field direction. These estimates are present in the text. Yes, it potentially can be q-par or nonstationary. Though, the probability to convert q-perp configuration to q-par one with an arbitrary rotation is small, because of the relatively narrow target solid angle.

*Conclusions: "shock structure should become independent from magnetic field direction" - how did you arrive at this conclusion ?*

In our mind, it is an evident logical extrapolation to infinitesimal upstream magnetic field.

*The above comments are also addressed to Discussion and Conclusions where many of the earlier statements are simply reiterated.*
*The paper also claims that "overall layout is quite characteristic and distinctly different from that for supercritical quasi-perpendicular shocks." It is not explained what is "characteristic" (see e.g. Scudder et al, 1986 as an example of how a "typical" shock should be described).*

Yes, Sec.4 naturally contains some reiteration and also mostly additional discussion of relevant issues. Some parts of it and Conclusions are now reworded. Under "characteristic" we mean certain similarity of all 22 considered shocks.
The main goal of this paper is consideration of large-amplitude variations, which are definitely different from that at q-perp supercritical shock. Scudder et al papers are nice, but do not contain sufficient waveform analysis.

*English is poor, please edit.*

OK

**Editor comments**

*Figure 2 i*s corrected (upper limit increased). There is no sense to add beta>20 line at the other panels, since there is almost no difference there for beta>10,20, while three lines are not easy to view on histograms.

*Yes, Fig 3 is not a shock crossing example*, just a sample solar wind

"*Therefore the presence of '2-Hz' waves might be due to some temporal shock front evolution*" – moved to discussion.

*In disc subsec 2 "In this study we advance our knowledge, determining some quantitative features of those variations." Describe how .*

This phrase is deleted to avoid confusion. Actually everything is explained in the next subsection.

*In Conclusions. However at some (very large) $\beta$ shock structure should become independent from magnetic field direction. MEANING*

We believe it is an evident logical statement. When magnetic field becomes very small and unimportant, some new physics in shock transition should appear. Now Conclusions are reworded.

[revised manuscript text omitted]

---

## Author Response (AR3)

I Would like to thank The Editor and Referees for their attention. There were no changes requested